# Single-cell analysis identifies the CNP/GC-B/cGMP axis as marker and regulator of modulated VSMCs in atherosclerosis

Moritz Lehners [1], Hannes Schmidt[1], Maria T. K. Zaldivia[1], Daniel Stehle [1], Michael Krämer[1], Andreas Peter [2], Julia Adler[3], Robert Lukowski [3], Susanne Feil [1] & Robert Feil [1] ✉

A balanced activity of cGMP signaling contributes to the maintenance of cardiovascular homeostasis. Vascular smooth muscle cells (VSMCs) can generate cGMP via three ligand-activated guanylyl cyclases, the NO-sensitive guanylyl cyclase, the atrial natriuretic peptide (ANP)-activated GC-A, and the C-type natriuretic peptide (CNP)-stimulated GC-B. Here, we study natriuretic peptide signaling in murine VSMCs and atherosclerotic lesions. Correlative profiling of pathway activity and VSMC phenotype at the single-cell level shows that phenotypic modulation of contractile VSMCs to chondrocyte-like plaque cells during atherogenesis is associated with a switch from ANP/GC-A to CNP/GC-B signaling. Silencing of the CNP/GC-B axis in VSMCs results in an increase of chondrocyte-like plaque cells. These findings indicate that the CNP/GC-B/cGMP pathway is a marker and atheroprotective regulator of modulated VSMCs, limiting their transition to chondrocyte-like cells. Overall, this study highlights the plasticity of cGMP signaling in VSMCs and suggests analogies between CNP-dependent remodeling of bone and blood vessels.

The second messenger 3′,5′-cyclic guanosine monophosphate (cGMP) is important for the maintenance of cardiovascular homeostasis in mammals including humans[1–4]. cGMP is generated from GTP by guanylyl cyclases and is degraded to 5′-GMP by phosphodiesterases (PDEs). A classic function of cGMP is the regulation of blood flow via relaxation of smooth muscle cells (SMCs), which leads to vasodilation. There is increasing evidence that cGMP also influences the growth and survival of cells in the vasculature, such as endothelial cells (ECs), vascular SMCs (VSMCs), and pericytes, thereby affecting vascular remodeling under disease conditions such as atherosclerosis[5–8]. In blood vessels, at least three members of the guanylyl cyclase family are expressed: the cytosolic nitric oxide (NO)-sensitive guanylyl cyclase (NO-GC, also known as soluble GC) and two transmembrane guanylyl cyclases, GC-A and GC-B, which are activated by natriuretic peptides[9–11]. GC-A (also known as Npr1 or Npr-A) is stimulated by atrial

natriuretic peptide (ANP) and B-type natriuretic peptide (BNP). GC-B (also known as Npr2 or Npr-B) is activated by C-type natriuretic peptide (CNP). Natriuretic peptides can also bind to the so-called clearance receptor Npr-C (also known as Npr3) that lacks guanylyl cyclase activity. Npr-C was first described for removing natriuretic peptides from the blood stream, but CNP-bound Npr-C might also have a cGMP-independent signaling function[12]. Many effects of cGMP on the vasculature, particularly on VSMCs, are mediated by the cGMP-dependent protein kinase type I (cGKI, gene name: *Prkg1*), which phosphorylates various target proteins[13].

The central role of natriuretic peptide signaling is illustrated by the high number of genetic variants of members of these pathways (e.g., ANP, BNP, GC-A, Npr-C) that are associated with cardiovascular diseases in humans[14–17]. The CNP/GC-B axis is well known for its role in the regulation of bone growth, and genetic variants of GC-B have been

[1]Interfakultäres Institut für Biochemie, University of Tübingen, Tübingen, Germany. [2]Department for Diagnostic Laboratory Medicine, Institute for Clinical Chemistry and Pathobiochemistry, University Hospital Tübingen, Tübingen, Germany. [3]Department of Pharmacology, Toxicology and Clinical Pharmacy, Institute of Pharmacy, University of Tübingen, Tübingen, Germany. ✉e-mail: robert.feil@uni-tuebingen.de

linked to short stature or skeletal overgrowth in humans[18]. In addition, the CNP/GC-B/cGMP pathway controls mammalian oocyte meiosis[19] and distinct functions of the somatosensory system[20]. Drugs that target the natriuretic peptide signaling system are successfully used in the clinic and are constantly being developed further[21,22]. For instance, a combination drug of valsartan, an angiotensin receptor blocker, with sacubitril, an inhibitor of the neutral endopeptidase neprilysin, is used to treat heart failure[23]. As the ectoenzyme neprilysin degrades natriuretic peptides, its inhibition increases natriuretic peptide-dependent cGMP generation. Recently, the CNP analog vosoritide was shown to be effective for the treatment of achondroplasia in short-limbed children[24]. In addition to its many beneficial effects, the CNP/GC-B/cGMP pathway may also promote the growth of tumors such as melanoma[25].

The ANP/GC-A system is well known for its role in blood pressure regulation. ANP is mainly secreted from the atrium. Its release can be increased by muscle stretch or neurohormones, and it acts mainly in an endocrine manner[26]. Activation of GC-A by ANP lowers blood pressure via several mechanisms including natriuresis and vasodilation. Studies using knockout mice have shown that GC-A in ECs and microcirculatory pericytes is important for chronic blood pressure control, while activation of GC-A in VSMCs contributes to the relaxation of larger arteries and acute blood pressure changes[27–29].

Compared to the ANP/GC-A system, the cardiovascular functions of CNP/GC-B signaling are less well understood. In contrast to ANP, CNP acts primarily in a para- or autocrine manner[30,31]. CNP release from ECs, for instance, appears to be regulated by its expression, which can be increased by inflammatory stimuli or shear stress[32]. Expression of GC-B has been reported in VSMCs and pericytes of blood vessels[33,34]. Global deletion of CNP or GC-B in mice leads to dwarfism and a significantly shortened lifespan[35,36]. This may mask less severe phenotypes and hinders long-term analysis of blood pressure or atherosclerosis in knockout animals. Another complicating factor is that effects of CNP might be mediated not only by GC-B but also by the "clearance receptor" Npr-C. Indeed, it has been shown that the pro-angiogenic effect of CNP depends on Npr-C but not GC-B[37]. Findings using global and cell type-specific mouse mutants led to an ongoing debate as to whether the blood pressure-lowering effect of endothelial CNP[33,38] is mediated via GC-B in microcirculatory pericytes[34] or Npr-C[38–40].

Many studies have reported effects of cGMP on vascular remodeling associated with atherosclerosis, but exactly how cGMP affects disease development at the cellular and molecular level is not clear[7,8]. ANP and CNP are considered to be atheroprotective. Cell culture studies reported anti-migratory and anti-proliferative effects of ANP and CNP on VSMCs[41–45]. Global genetic ablation of GC-A in mice promoted atherosclerosis, but it is not clear whether this phenotype was due to the hypertension of GC-A knockout mice and/or the lack of a local protective effect of GC-A in atherosclerotic lesions[46]. EC-specific deletion of CNP also accelerated the development of atherosclerotic lesions in mice[38]. It is unlikely that the atheroprotective effect of CNP is mediated by Npr-C, as global or EC-specific ablation of Npr-C attenuated the development of atherosclerotic plaques, indicating that Npr-C promotes atherogenesis[47]. The existing literature thus suggests that CNP attenuates atherogenesis, possibly via stimulation of GC-B and increased cGMP signaling in the vessel wall, while Npr-C acts as a clearance receptor for CNP counteracting atheroprotection by CNP. However, whether genetic ablation of GC-B influences atherosclerosis has not been reported.

During the course of atherosclerosis, quiescent VSMCs in the vascular media begin to change their phenotype by losing their contractile properties and transitioning to a cell state characterized by proliferation, migration and extracellular matrix synthesis[48,49]. This transition from contractile to synthetic/dedifferentiated VSMCs is referred to as phenotypic modulation. Given the phenotypic heterogeneity of VSMCs in the atherosclerotic vessel wall, we hypothesized that cGMP signaling pathways may differ between contractile and modulated VSMCs[8]. To test this hypothesis and study cGMP signaling heterogeneity in VSMCs, methods that can resolve the expression, or better functional activity, of the various cGMP pathways at the single-cell level need to be used. In most previous studies, the expression of GC-A or GC-B was analyzed using bulk techniques that do not provide cellular resolution, such as Western blotting, Northern blotting, or qPCR. One reason for the apparent difficulty in detecting GC-A or GC-B at the single-cell level is the fact that these transmembrane receptors are expressed at presumably low levels, which makes their detection with antibodies on tissue sections a major challenge. In fact, only a few studies have reported antibody-based detection of GC-A or GC-B at the cellular level (e.g., refs. 50,51). Recent developments in the field of fluorescent cGMP biosensors have led to highly sensitive tools for detecting the enzymatic activity of guanylyl cyclases including GC-A and GC-B in living cells[52,53]. These biosensors enable the detection of cGMP in real time with single-cell resolution in populations of cultured cells. In addition, this technology allows the analysis of cGMP signals in blood vessels of transgenic cGMP sensor mice under close-to-native conditions[54–57].

In the present study, we characterized natriuretic peptide signaling pathways in VSMCs at the single-cell level, focusing on the CNP/GC-B/cGMP axis. We monitored and manipulated the activity of the pathways using a combination of real-time cGMP imaging in cultured VSMCs and isolated arteries, immunostaining, single-cell RNA sequencing (scRNA-seq), and genetic approaches. We show that functional ANP/GC-A and CNP/GC-B pathways are predominantly expressed in contractile and modulated VSMCs, respectively, and we identify the CNP/GC-B/cGMP axis as a marker and modulator of dedifferentiated chondrocyte-like VSMCs in atherosclerosis.

## Results

### Different VSMC phenotypes show different cGMP responses

We used cultures of primary mouse aortic VSMCs as a model to compare cGMP generation between individual VSMCs with different phenotypes. When cultured in vitro, VSMCs convert from a contractile/differentiated state to a synthetic/dedifferentiated state[48]. This phenotypic switch is associated with a characteristic morphological change from elongated to rhomboid cells with reduced expression of markers for contractile VSMCs such as α-smooth muscle actin (αSMA) and smooth muscle protein 22-α (SM22α). Similar changes likely occur in vivo during the formation of atherosclerotic lesions[49].

Western blot analysis confirmed that our primary VSMCs expressed GC-B and NO-GC (Fig. 1a). Expression of the GC-A protein could not be analyzed for lack of suitable antibodies. To investigate the functional expression of cGMP signaling pathways in individual VSMCs, we prepared VSMC cultures from aortae of transgenic cGMP sensor mice[54] that express a fluorescence resonance energy transfer (FRET)-based biosensor for cGMP, cGi500 (cGMP indicator with an apparent $EC_{50}$ of 500 nM)[58]. The intracellular cGMP concentrations of cells were measured in real time by FRET microscopy of subconfluent cultures stimulated with ANP, CNP, or the NO donor DEA/NO (hereafter referred to as "NO") (Fig. 1b–d). Binding of cGMP to the biosensor leads to an increase in CFP fluorescence and a concomitant decrease in YFP fluorescence (Fig. 1b, cyan and yellow traces, respectively). Thus, changes in the ratio of CFP over YFP fluorescence reflect changes in the cGMP concentration over time (Fig. 1b, black ratio traces, R - [cGMP]). As expected from previous cGMP imaging experiments in VSMCs[54,59,60], single-cell cGMP/FRET measurements showed robust cGMP responses to ANP, CNP and/or NO in almost all cells (Fig. 1b). Only ≈7% of the measured cells showed no response to any of these stimuli or could not be evaluated for technical reasons.

To compare the response of individual VSMCs to ANP vs. CNP, we sequentially stimulated the same cells with 50 nM and 250 nM of the

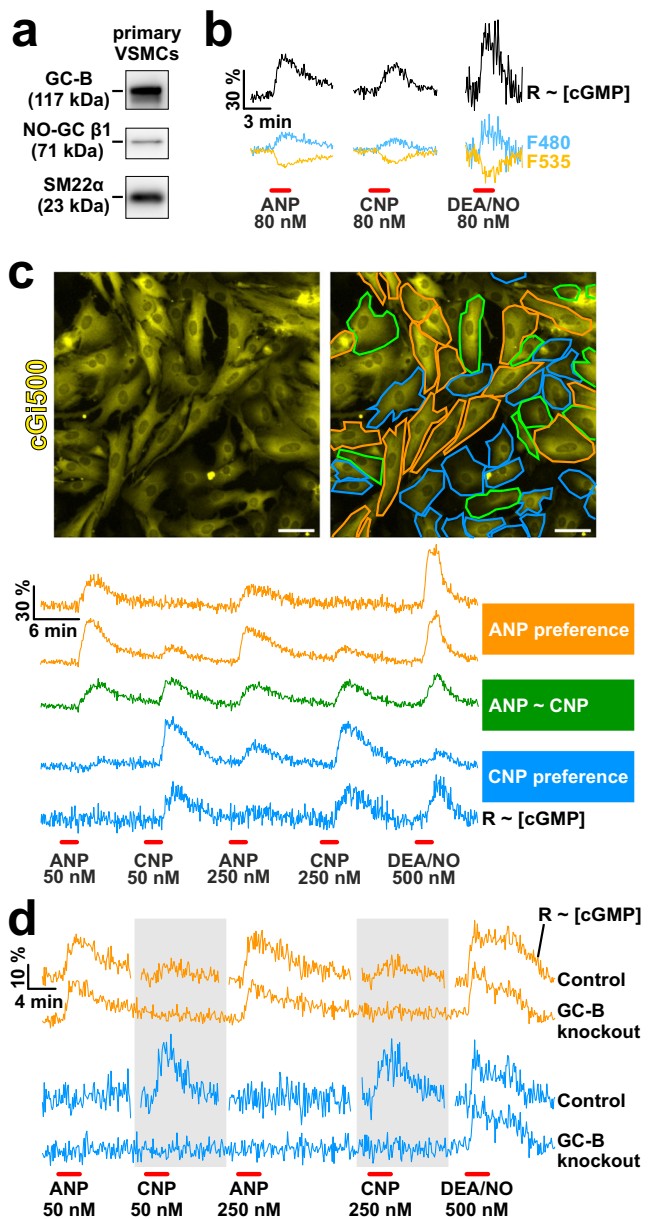

**Fig. 1 | Single-cell analysis of cGMP signaling diversity in primary VSMCs by real-time cGMP/FRET imaging. a** Western blot analysis of VSMC lysates. Applied antibodies and expected molecular weights of the target proteins are indicated. One representative blot out of three independent experiments is shown. **b** cGMP/FRET imaging of primary VSMCs isolated from global cGMP sensor mice. VSMCs were stimulated with (left) ANP, (middle) CNP, or (right) DEA/NO (80 nM each, red bars). Cyan and yellow traces show CFP and YFP fluorescence of the sensor, respectively. Black traces indicate the intracellular cGMP concentration over time (ratio trace of CFP/YFP or R - [cGMP]); shown are means of all cells that reacted to the respective stimulus (left: 20 analyzed out of 26 recorded cells; middle: 33 out of 35 cells; right: 42 out of 57 cells; one coverslip each from the same cell isolation). The scale bars indicate the time and percent change of the traces relative to baseline. **c** Representative cGMP/FRET measurements (ratio traces R - [cGMP]) of individual VSMCs that were consecutively stimulated with ANP, CNP, and DEA/NO (red bars, concentrations indicated in the panel). Based on the analysis of 438 out of 516 recorded cells (on three coverslips from one cell isolation), the cells were classified as "ANP-preferring" (orange, 124 cells), cells without a clear preference for ANP or CNP ("ANP - CNP", green, 65 cells), and "CNP-preferring" (cyan, 249 cells). The black scale bars indicate the time and percent change of the traces relative to baseline. The pictures on top show the cGMP sensor expressing VSMCs visualized by the YFP fluorescence of cGi500. In the right picture, the cGMP response pattern of each cell is highlighted using the same color code as for the cGMP traces in the lower panel. White scale bars, 50 µm. **d** Comparison of cGMP responses in VSMCs from control and global GC-B knockout mice. cGMP/FRET measurements (ratio traces R - [cGMP]) were performed as in (**c**). The responses are representative of "ANP-preferring" (orange) and "CNP-preferring" (cyan) cells. Shown are means from one representative measurement per genotype (from top to bottom: 55 cells, 68 cells, 6 cells, 25 cells). For the control measurement, ratio traces for 61 out of 87 recorded cells (on one of two coverslips from one cell isolation) are shown. Cells classified as ANP - CNP cells are not shown and, therefore, not included in the total count of recorded cells. For the GC-B knockout measurement, ratio traces for 93 out of 137 recorded cells (on one of two coverslips from one cell isolation) are shown. As the control measurement lasted longer, some data points from the baseline were omitted (gaps in the control traces) to align the drug applications without distorting the time axis. The scale bars indicate the time and percent change of the traces relative to baseline. These results were confirmed by an independent experiment with GC-B knockout VSMCs that were transfected with the cGi500 biosensor. As detailed in the Methods section, cells showing a poor quality of their ratio traces were excluded from analysis. The respective numbers of analyzed cells out of total recorded cells are indicated above. Source data including exact p-values and applied statistical tests are provided in the Source Data file.

peptides, followed by a control stimulation with NO at the end of each experiment (Fig. 1c). In general, the cGMP peaks were similar upon stimulation with 50 nM and 250 nM of the peptides, indicating that 50 nM of ANP or CNP were sufficient to fully activate their respective receptors, GC-A and GC-B, on the VSMCs. Using this approach, we identified cell-to-cell heterogeneity of natriuretic peptide-dependent cGMP signaling in primary aortic VSMCs. Inspection of single-cell cGMP measurements revealed three response patterns (Fig. 1c, Supplementary Movie 1): (1) cells that responded much stronger to ANP than to CNP ("ANP-preferring cells", orange); (2) cells without a clear preference for ANP or CNP ("ANP - CNP cells", green); and (3) cells that responded much stronger to CNP than to ANP ("CNP-preferring cells", cyan). Note that from "ANP-preferring" to "ANP - CNP" to "CNP-preferring" cells, cGMP responses to ANP decrease, while cGMP responses to CNP increase. Detailed analysis of the single-cell cGMP imaging data revealed that the cells of a VSMC population form a continuous gradient with respect to their preference for natriuretic peptides (Supplementary Fig. 1a). To confirm that CNP-induced cGMP signals were indeed generated via GC-B, we analyzed GC-B knockout VSMCs (Fig. 1d). These cells responded normally to ANP and NO, showing that

the respective cGMP-generating systems were not impaired in the absence of GC-B. Their responsiveness to ANP indicated that the cells expressed GC-A, which in principle could also mediate CNP-induced cGMP increases. However, GC-B knockout VSMCs did not respond to CNP at all, demonstrating that CNP-induced cGMP generation was exclusively mediated by GC-B (Fig. 1d).

Interestingly, the morphology of most ANP-preferring VSMCs was elongated, whereas CNP-preferring cells appeared to be more roundish (Fig. 1c, right picture, orange and cyan cell borders, respectively). Indeed, quantitative analysis of cell morphology revealed significant increases of roundness from ANP-preferring over ANP - CNP to CNP-preferring cells (Supplementary Fig. 1b). These morphological differences suggested that ANP and CNP preference might be associated with the phenotype of contractile and modulated VSMCs, respectively. To test this hypothesis, we used a mapping method in which VSMCs are grown on gridded coverslips (Supplementary Fig. 2). First, we determined the cells' cGMP response patterns by live-cell FRET imaging as exemplified in Fig. 1c. We then fixed the cells and stained them for marker proteins by immunofluorescence. This procedure allowed us to directly correlate the cGMP response pattern of a given cell with its phenotypic state defined by immunostaining. We stained αSMA and SM22α as marker proteins for contractile VSMCs[49] (Fig. 2a) and the non-SMC proteins S100A4 and platelet-derived growth factor receptor α (PDGFRα), which indicate modulated VSMCs[61–64].

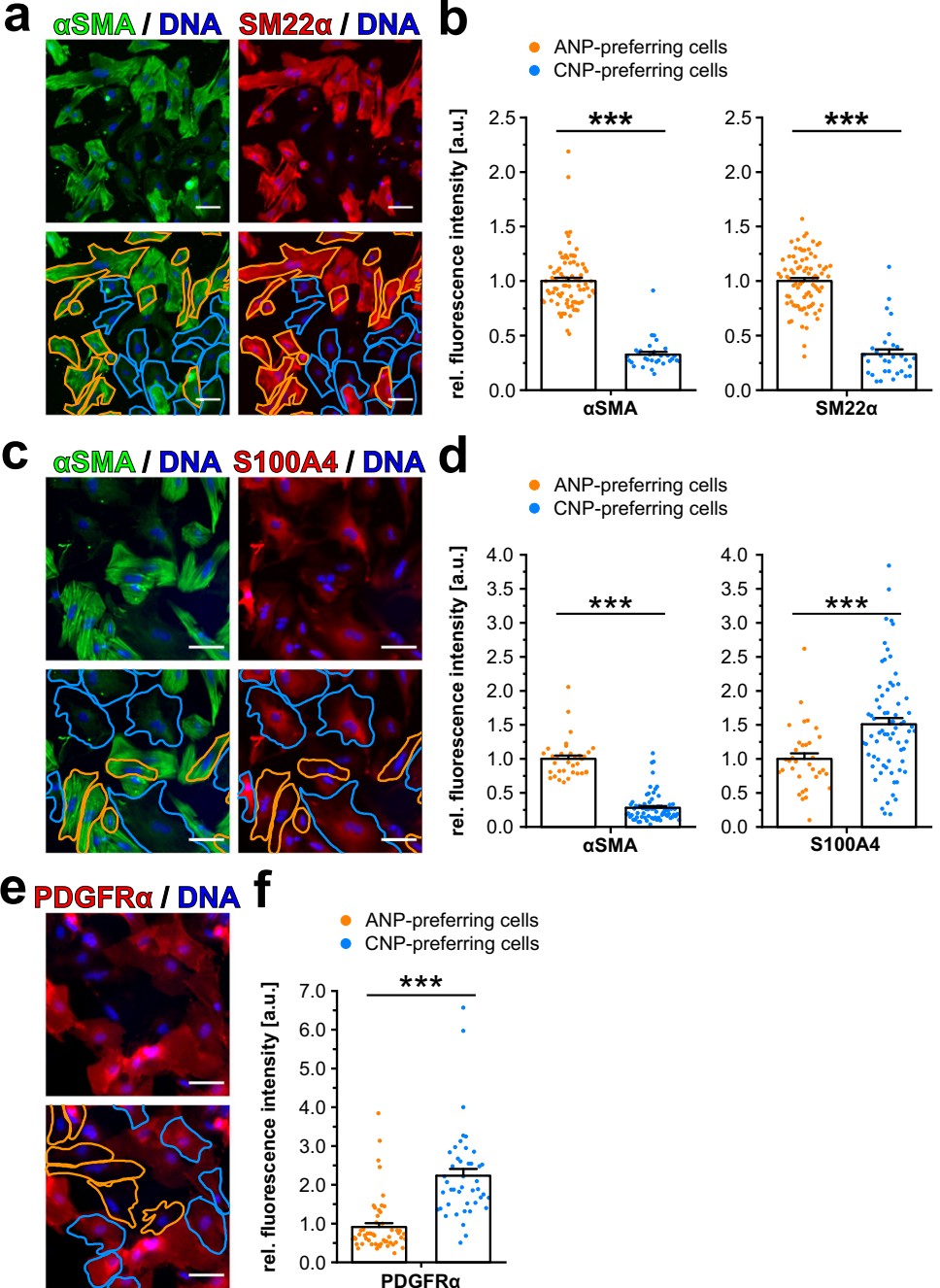

**Fig. 2 | Correlation of VSMC phenotype and cGMP response pattern at the single-cell level.** To correlate the phenotype and cGMP response pattern of an individual cell, we have used a mapping method based on gridded coverslips. For details, see Methods section and Supplementary Fig. 2. Primary VSMCs from global cGMP sensor mice were grown and imaged for cGMP as described in Fig. 1, followed by immunofluorescence staining for marker proteins. The panels show representative images and the quantitative evaluation of fluorescence intensities (normalized to the mean of ANP-preferring cells) of individual VSMCs stained for **a**, **b** contractile markers αSMA and SM22α, **c**, **d** αSMA and the non-SMC protein S100A4 and **e**, **f** the non-SMC protein PDGFRα. ANP-preferring cells are indicated by orange cell borders and data points, and CNP-preferring cells by cyan cell borders and data points. Each data point represents an individual VSMC. Data are shown as mean + SEM. **b** 112 cells were analyzed out of 114 recorded cells on two coverslips

from one cell isolation. **d** 105 cells were analyzed out of 113 recorded cells on four coverslips from one cell isolation. **f** 97 cells were analyzed out of 115 recorded cells on five coverslips from one cell isolation. Statistical significance is indicated by asterisks (***$p < 0.001$). Scale bars, 50 μm. DNA was stained with Hoechst No. 33258 (blue). Similar results were obtained in another independent experiment. As detailed in the Methods section, cells that showed a poor quality of their ratio traces or that could not be analyzed in the immunostaining due to limitations of the mapping technique were excluded from analysis. The respective numbers of analyzed cells out of total recorded cells are indicated above. Cells classified as ANP - CNP cells are not shown and, therefore, not included in the total count of recorded cells. Source data including exact p-values and applied statistical tests are provided in the Source Data file.

Consistent with their morphology, ANP-preferring cells expressed significantly higher levels of αSMA and SM22α than CNP-preferring cells (Fig. 2a, b). Conversely, CNP-preferring cells expressed higher amounts of S100A4 and PDGFRα than ANP-preferring cells (Fig. 2c–f). The increased expression of S100A4 in CNP-preferring cells was accompanied by a marked decrease in αSMA expression in the same cells (Fig. 2c, lower pictures, cells with cyan cell borders).

The VSMC cultures used for the experiments described above were isolated from transgenic mice with ubiquitous expression of the cGMP sensor. Therefore, it would have been possible that our cGMP imaging data were "contaminated" by cells of non-SMC origin that might have been present in the cell cultures and expressed the cGMP sensor. To rule out this possibility, we performed cGMP/FRET imaging experiments with VSMC cultures prepared from aortae of SMC-specific cGMP sensor mice. In these experiments, we observed a very similar correlation between cell phenotype and cGMP response pattern as in VSMC cultures from global sensor mice (Supplementary Fig. 3). Thus, we conclude that our cGMP/FRET data were not confounded by non-VSMCs. Some of the VSMCs appeared to have reduced expression of SMC markers due to phenotypic modulation during culture. Taken together, correlative cGMP imaging and immunostaining of marker proteins at the single-cell level demonstrated a marked heterogeneity of natriuretic peptide-induced cGMP signaling in VSMCs, which depends on the phenotypic state of the cells. While contractile VSMCs predominantly expressed the ANP/GC-A axis, modulated VSMCs were characterized by pronounced CNP/GC-B signaling.

Next, we profiled the mRNA expression of our primary aortic VSMCs by scRNA-seq and identified three phenotypic states, namely contractile, proliferating, and modulated SMCs (cSMCs, pSMCs, and mSMCs, respectively), as well as a small cluster of endothelial cells (ECs, ≈2% of all cells) (Fig. 3a, b). Compared to p/mSMCs, cSMCs expressed higher levels of markers for contractile SMCs, such as smooth muscle myosin heavy chain (gene name: *Myh11*), αSMA (*Acta2*), or SM22α (*Tagln*) (Fig. 3b, c left panels with red dots). The p/mSMCs expressed high levels of markers for modulated/dedifferentiated SMCs like lumican (*Lum*), PDGFRα, or S100A4 (Fig. 3b, c middle panels with blue dots). This mRNA expression data correlated well with the expression of the respective proteins detected by immunofluorescence staining of our VSMC cultures (Fig. 2). As expected from previous bulk mRNA analysis of primary VSMCs by RT-PCR[54], scRNA-seq detected the expression of NO-GC, GC-A, and GC-B at the single-cell level (Fig. 3c right panels with green dots). Consistent with our cGMP/FRET measurements, p/mSMCs expressed higher levels of the CNP receptor GC-B (*Npr2*) than cSMCs (Fig. 3c, d). The mRNA encoding the α2 subunit (*Gucy1a2*) of the NO-GC 2 isoform was detected in only a few cells (Fig. 3d), while the subunits forming the NO-GC 1 isoform (*Gucy1a1* and *Gucy1b1*) were strongly expressed in cSMCs and downregulated in p/mSMCs (Fig. 3c, d). The downregulation of NO-GC in modulated VSMCs detected by scRNA-seq was confirmed by single-cell immunostaining and cGMP/FRET imaging, which demonstrated a significant reduction of the NO-GC β1 subunit and NO-induced cGMP signals in CNP-preferring αSMA^low cells compared with ANP-preferring αSMA^high cells (Supplementary Fig. 4). The mRNA encoding the ANP receptor GC-A (*Npr1*) was predominantly expressed in ECs and was only very weakly detectable in VSMCs, albeit with a preference in cSMCs over p/mSMCs (Fig. 3c, d). The difficulty in detecting GC-A in VSMCs at the mRNA level by scRNA-seq contrasts with the robust ANP-dependent cGMP responses that we observed in ≈63% of our primary VSMCs (see, ANP-preferring and ANP ~ CNP cells in Figs. 1, 2 and 4, and in Supplementary Figs. 3 and 4). These findings support the notion that live-cell cGMP/FRET imaging of the activity of a cGMP-generating guanylyl cyclase such as GC-A can be more sensitive to assess the functional relevance of the pathway than its detection at the mRNA or protein level. Furthermore, the scRNA-seq data showed that our primary VSMCs expressed, in addition to the aforementioned

cGMP generators, the natriuretic peptide clearance receptor Npr-C (*Npr3*), the cGMP effector cGKI (*Prkg1*), and the cGMP degraders *Pde1a*, *Pde3a* and *Pde5a*, while mRNA levels for *Prkg2*, *Pde2a*, *Pde9a* and *Pde10a* were low to undetectable (Fig. 3d). Interestingly, the expression profile of GC-B (*Npr2*) appeared to overlap extensively with VSMCs expressing osteoprotegerin (*Tnfrsf11b*) (Fig. 3e), a marker of so-called chondromyocytes, which are VSMC-derived chondrocyte-like cells found in atherosclerotic plaques[65]. Indeed, ≈24% of the mSMCs co-expressed *Npr2* and *Tnfrsf11b* mRNA (Fig. 3e yellow dots).

Together, our combined single-cell analyses using cGMP/FRET imaging, immunostaining, and scRNA-seq revealed a marked heterogeneity and plasticity of natriuretic peptide-dependent cGMP signaling in VSMCs in vitro. Our findings strongly suggested that phenotypic modulation of VSMCs results in downregulation of cGMP signaling via the ANP/GC-A pathway and upregulation of the CNP/GC-B axis in VSMC-derived cells, including chondrocyte-like cells.

### Development of CNP preference parallels VSMC phenotypic modulation

Because phenotypic conversion to modulated cells is a dynamic process that evolves over time, we tracked the expression of GC-B at the single-cell level during growth of VSMC cultures. Since no suitable antibodies for immunostaining of GC-B in VSMCs were available, we used VSMCs from GC-B LacZ reporter mice. These mice carry a LacZ transgene integrated into the endogenous GC-B gene locus so that β-galactosidase containing a nuclear localization signal is expressed from one of the two GC-B alleles instead of GC-B[66]. Thus, cells actively transcribing the GC-B gene can be conveniently detected by staining β-galactosidase activity with X-gal (Fig. 4a). While we detected virtually no X-gal⁺ cells at the beginning of cultivation, there was a significant increase in the percentage of X-gal⁺ cells between 4 and 7 days of culture under standard conditions in the presence of 10% FCS (Fig. 4b). Next, VSMCs were grown under conditions known to attenuate (low serum concentrations) or promote (high serum concentrations, repeated passaging) phenotypic modulation[48,67], and then the cGMP signaling phenotype was assessed by cGMP/FRET imaging. Compared to standard culture conditions, i.e., growth of primary VSMCs (p0) in 10% FCS, a reduction of the serum concentration led to a marked decrease of CNP-preferring cells, whereas an increase in serum concentration or passaging of the primary VSMCs resulted in a dramatic increase of CNP-preferring cells (Fig. 4c, d). The passaging-induced increase in VSMCs with a functionally active CNP/GC-B/cGMP axis (Fig. 4d) was associated with an increase of GC-B protein expression as detected by Western blot analysis (Fig. 4e).

Overall, these results strongly supported the conclusion that the CNP/GC-B/cGMP axis is a marker for modulated VSMCs. It is important to note that the proportion of GC-B⁺ cells detected by the LacZ reporter approach (≈3% after 7 days in culture, Fig. 4b) was much smaller than the fraction of CNP-preferring cells detected by cGMP/FRET imaging under similar growth conditions (Fig. 4c, d). This discrepancy indicates that expression of the β-galactosidase reporter enzyme under control of the GC-B promoter was too low in most cells to be detected by X-gal staining. Since the genetic GC-B LacZ reporter strategy appeared to detect only cells with relatively strong expression of the GC-B gene, we refer to the X-gal⁺ cells also as GC-B^high cells. On the other hand, we were able to detect endogenous GC-B enzymatic activity with high sensitivity by visualizing CNP-induced cGMP signals with the FRET-based cGMP biosensor.

### Atherosclerotic arteries show robust CNP-induced cGMP signals

Our previous results were obtained in a cell culture model in vitro. It was therefore important to investigate the significance of CNP-dependent cGMP signaling in VSMCs under conditions that more closely resemble the in vivo situation. Based on our cell culture experiments, we hypothesized that phenotypic modulation of VSMCs

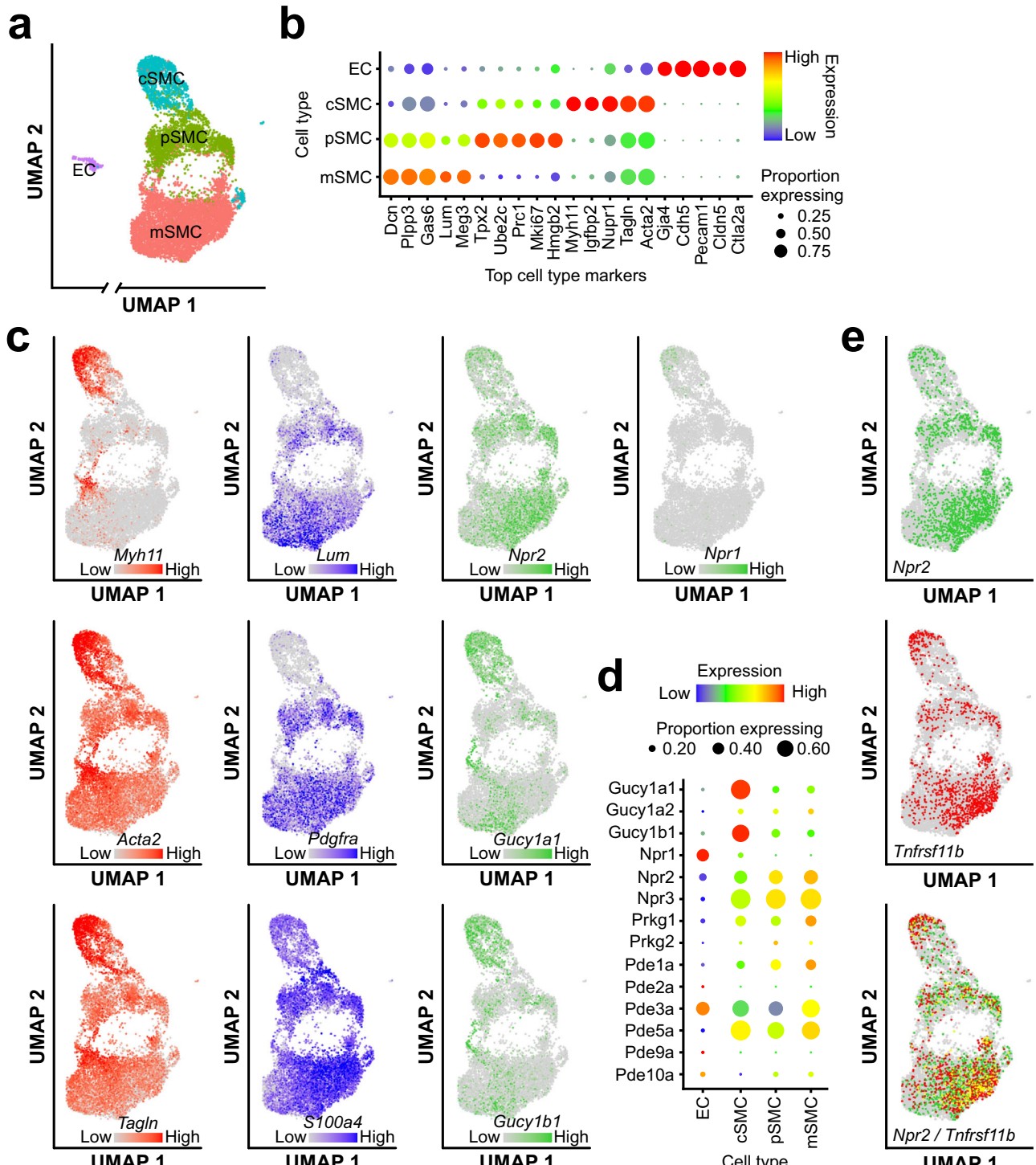

**Fig. 3 | Transcriptomic characterization of primary VSMCs by scRNA-seq.**
**a** Uniform Manifold Approximation and Projection (UMAP) visualization of the phenotypic diversity of primary aortic SMCs after 5 days in culture, demonstrating the coexistence of contractile and modulated SMCs. Each dot represents a single cell. Cell types represented by each cluster are indicated: EC endothelial cells, cSMC contractile SMCs, pSMC proliferating SMCs, mSMC modulated SMCs. **b** Dot plot showing the expression of the top five genes defining each cell cluster shown in (**a**). **c** UMAP visualization of the expression of contractile SMC marker genes (left, red), non-SMC marker genes (middle, blue), and cGMP generator genes (right, green) in the same cells as shown in panel a. The expression level is illustrated by a color scale. Each dot represents a single cell. For cell type identity, see (**a**). **d** Dot plot showing the expression of cGMP pathway genes in each cluster/cell type. The size of each dot represents the fraction of cells per cluster with at least one transcript for the respective gene detected and the color scale indicates the expression level of the respective gene. **e** UMAP visualization of the co-expression of Npr2 and Tnfrsf11b in the cells shown in panel a. Cells with high expression of Npr2 (green), Tnfrsf11b (red), and co-expression of Npr2 and Tnfrsf11b (yellow) are shown. Cells in which at least one transcript of both genes was detected were classified as co-expressing cells.

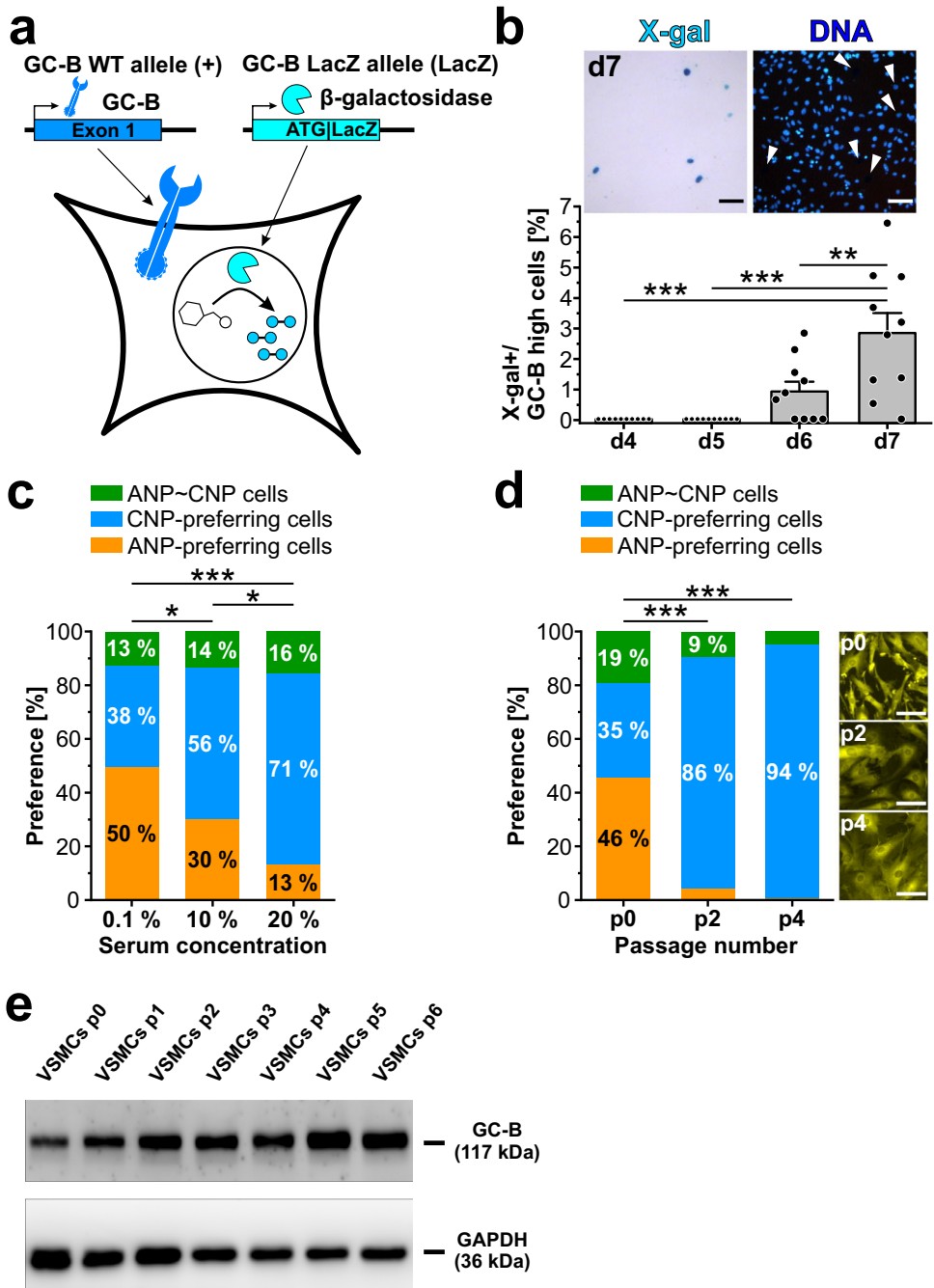

in vivo, as it occurs during the progression of atherosclerosis, leads to the development of CNP-responsive cells. To test this, we determined cGMP signaling profiles in healthy and atherosclerotic arteries of cGMP sensor mice by ex vivo cGMP/FRET measurements. Segments of isolated aortae or smaller arteries, such as the carotid or brachiocephalic artery, were opened longitudinally and imaged from the luminal side with a confocal spinning disk microscope (Fig. 5a). With this setup, it was possible to identify VSMCs by their morphology and to measure cGMP/FRET signals in individual cells. However, only the outer cell layers of an atherosclerotic plaque (presumably the fibrous cap, shoulder, and upper region of the plaque core), but not the inner core region, could be measured with sufficient intensity to evaluate cGMP signals (Fig. 5a green plane in right image; and Fig. 5c left image).

Real-time cGMP/FRET measurements revealed different cGMP response patterns in healthy and atherosclerotic arteries. Healthy arteries showed robust cGMP signals triggered by ANP and NO (Fig. 5b, d), like ANP-preferring contractile VSMCs in culture. In healthy arteries, we did not detect CNP-induced cGMP increases. In contrast, CNP potently increased cGMP in cells of atherosclerotic plaques, as did ANP and NO (Fig. 5c, d, Supplementary Movie 2). Plaque areas with strong cGMP responses to all three stimuli (CNP, ANP, and NO) were detected in ≈22% of all regions analyzed in atherosclerotic lesions. Consistent with the downregulation of NO-GC in modulated VSMCs in cell culture (Fig. 3d, Supplementary Fig. 4), we found significantly fewer NO-responsive regions in atherosclerotic arteries than in healthy blood vessels (Fig. 5d). The most striking difference between healthy and atherosclerotic arteries in terms of cGMP signaling was the almost exclusive presence of the CNP/GC-B axis in diseased vs. healthy vessels (Fig. 5d). The CNP-responsive plaque regions mirrored the signaling behavior of ANP ~ CNP cells and CNP-preferring cells present in VSMC cultures. Interestingly, ≈8% of measured plaque regions responded to

**Fig. 4 | Development of CNP/GC-B-dependent cGMP signaling during VSMC culture and promotion of CNP-preferring cells by phenotypic modulation.**
**a** Working principle of the GC-B LacZ reporter gene. X-gal staining of cells from GC-B LacZ reporter mice leads to blue stained nuclei in cells that express the GC-B gene at the time of staining. Note that GC-B-dependent cGMP signaling in heterozygous GC-B LacZ reporter cells is ensured by the presence of the GC-B wildtype allele. Cyan handles in the nucleus indicate indigo dye that stains nuclei of GC-B expressing cells. **b** Development of GC-B expression in primary VSMCs grown for 4–7 days in the presence of 10% FCS. Cells with presumably high GC-B expression were identified by their X-gal-stained nuclei (X-gal + /GC-B high cells). Each data point represents the percentage of X-gal +/GC-B high cells in a random field of view. Data are shown as mean + SEM ($n = 10$ regions on two coverslips per time point from one cell isolation). Statistical significance vs. day 7 is indicated by asterisks (**$p < 0.01$; ***$p < 0.001$). Similar results were obtained in another independent experiment. The inset shows a representative region of cultured VSMCs at day 7. White arrowheads indicate the position of X-gal+ nuclei. DNA was stained with Hoechst No. 33258 (blue). Note that nuclei showing an intense X-gal staining (dark blue) were not visible in the Hoechst No. 33258 channel due to interference of the X-gal precipitate with Hoechst fluorescence. Scale bars, 100 μm. **c**, Effect of serum concentration on ANP/CNP preference of primary VSMCs. Cells were grown under standard conditions in cell culture medium supplemented with 0.1%, 10% or 20% FCS for 48 h before cGMP/FRET measurements. Drugs were applied as shown in Fig. 1c. Bars indicate the fraction of cells in each category under each condition

(0.1% FCS: 167 cells were analyzed out of 172 recorded cells on three coverslips from three cell isolations; 10% FCS: 96 cells were analyzed out of 98 recorded cells on two coverslips from two cell isolations; 20% FCS: 121 cells were analyzed out of 123 recorded cells on two coverslips from two cell isolations). **d** Effect of passaging on ANP/CNP preference of VSMCs. Primary and passaged VSMCs were grown under standard conditions in 10% FCS and analyzed by cGMP/FRET measurements. Bars indicate the fraction of cells in each category under each condition (p0: 284 cells were analyzed out of 291 recorded cells on five coverslips from two cell isolations; p2: 214 cells were analyzed out of 232 recorded cells on six coverslips from two cell isolations; p4: 123 cells were analyzed out of 167 recorded cells on five coverslips from two cell isolations). On the right, representative images of the VSMCs (visualized by the YFP fluorescence of cGi500) used for cGMP/FRET measurements are shown. Scale bars, 100 μm. In panels c and d, statistically significant differences in the distribution of ANP-preferring, ANP - CNP, and CNP-preferring cells under different conditions are indicated by asterisks (*$p < 0.05$; ***$p < 0.001$). As detailed in the Methods section, cells showing a poor quality of their ratio traces were excluded from analysis. The respective numbers of analyzed cells out of total recorded cells are indicated above. **e** Effect of passaging on GC-B expression in VSMCs detected by Western blotting. Applied antibodies and expected molecular weights of the target proteins are indicated on the left. Similar results were obtained in another independent experiment, where low passages (p1-p3) of VSMCs were compared. Source data including exact p-values and applied statistical tests are provided in the Source Data file.

CNP but not ANP, like the modulated VSMCs with strong CNP preference observed in cell culture.

The arteries used for the cGMP imaging experiments described above were isolated from mice with global sensor expression in all cells. Therefore, it was possible that our imaging data were confounded by non-SMCs that might have been present in the measured blood vessels and expressed the cGMP sensor, such as ECs. To rule this out, we performed cGMP/FRET imaging experiments with arteries from mice with SMC-specific expression of the cGMP sensor. In these experiments, we observed very similar cGMP response patterns as in the global sensor mice (Supplementary Fig. 5).

## SMC-specific GC-B ablation alters atherosclerosis

The comparison of healthy and atherosclerotic arteries by cGMP/FRET measurements revealed that the presence of a functional CNP/GC-B/ cGMP axis is a characteristic feature of atherosclerotic plaques. To unravel the specific role of GC-B-dependent cGMP signaling in VSMCs for atherosclerosis, we generated a genetic mouse model in which GC-B expression can be specifically abolished in SMCs in a temporally controlled manner (GC-B$^{smko}$ mice, Supplementary Fig. 6). In addition, both the control mice expressing GC-B as well as the GC-B$^{smko}$ mice carried the GC-B LacZ reporter allele. This allowed us to detect cells with active transcription of the GC-B gene by X-gal staining of plaques. Note that the LacZ reporter strategy identifies cells with an active GC-B promoter even in GC-B$^{smko}$ mice, in which GC-B protein expression was abolished (Supplementary Fig. 6). Postnatal SMC-specific ablation of GC-B was achieved with the Cre/loxP system using the tamoxifen-inducible αSMA-CreER$^{T2}$ mouse line[68]. All mice for these experiments were bred on an ApoE-deficient background and fed an atherogenic diet for 18 weeks before atherosclerosis was analyzed.

Western blot analysis confirmed that the GC-B protein was expressed in atherosclerotic aortae of control mice and efficiently deleted in GC-B$^{smko}$ mice without altering the levels of NO-GC and cGKI, which are also involved in vascular cGMP signaling (Fig. 6a). Physiological parameters and lipid profiles relevant to atherogenesis were similar in control and GC-B$^{smko}$ mice (Supplementary Table 1). Our GC-B$^{smko}$ mice showed neither the dwarfism nor the early mortality that is known for global GC-B knockout mice[36]. This is also in line with observations from Nakao and colleagues who showed that SMC-specific ablation of GC-B in mice does not affect blood pressure[33].

Based on Oil Red O staining, the total area of atherosclerotic lesions in the aorta was similar in control and GC-B$^{smko}$ mice (Fig. 6b, c).

In both groups, ≈20% of the aorta was covered by plaques (control: $21 \pm 2\%$ vs. GC-B$^{smko}$: $19 \pm 3\%$, n.s.; Fig. 6c), There were also no significant genotype-dependent differences when lesion size was analyzed separately in the aortic arch and thoracic/abdominal aorta of male and female mice (Supplementary Fig. 7). However, immunostaining of aortic sections for marker proteins and X-gal staining for GC-B promoter activity as a means of detecting modulated VSMCs suggested an altered plaque composition when GC-B was ablated in VSMCs. As shown in Fig. 6d, e, both genotypes had plaques covered by a characteristic αSMA$^+$ fibrous cap, and significantly less αSMA was found in the core region known to contain clonally-derived modulated VSMCs with low/absent αSMA expression[69–72]. There was a trend toward a decreased total αSMA$^+$ area in plaques from GC-B$^{smko}$ mice compared to control mice (GC-B$^{smko}$: $19 \pm 2\%$ vs. control: $27 \pm 4\%$, n.s.). Interestingly, this difference appeared to be due to a greater reduction of αSMA in the core of GC-B$^{smko}$ plaques compared to the core of control lesions (Fig. 6d, e). Indeed, the cap-to-core ratio of αSMA was significantly higher in GC-B$^{smko}$ plaques than in control lesions (Fig. 6f).

Next, we investigated if the reduction of αSMA in the plaque core regions of GC-B$^{smko}$ mice was due to enhanced phenotypic modulation of VSMCs in the absence of GC-B. Since our previous experiments have shown that GC-B expression marks modulated VSMCs, we used the GC-B LacZ reporter system to identify modulated VSMCs by X-gal staining of plaque sections. Indeed, we found X-gal$^+$ cells in atherosclerotic lesions of both control and GC-B$^{smko}$ mice (Fig. 6g). As expected from the relatively low sensitivity of LacZ reporter detection already noted in our cell culture experiments with GC-B LacZ VSMCs (see, Fig. 4b), the fraction of X-gal$^+$ cells detected in plaque sections (Fig. 6g) was much lower than the ≈41% of regions that showed CNP-induced cGMP responses in our cGMP/FRET measurements of atherosclerotic plaques (Fig. 5d). We observed in both control and GC-B$^{smko}$ lesions that most X-gal$^+$ cells accumulated in the plaque core and only few were found in the cap region (Fig. 6g). Interestingly, plaques of GC-B$^{smko}$ mice showed a strong trend toward more X-gal$^+$ cells compared with plaques of control mice (GC-B$^{smko}$: $5 \pm 2$ vs. control: $2 \pm 0.4$ X-gal$^+$ cells per positive section, n.s.) and they contained significantly more X-gal$^+$ cells than the media (Fig. 6h). Such a strong accumulation of X-gal$^+$ cells in the plaques compared with the media was not observed in atherosclerotic control mice (Fig. 6h). The increase of X-gal$^+$ cells together with the decrease in αSMA$^+$ area in plaques of GC-B$^{smko}$ mice suggested that silencing of CNP signaling in VSMCs by ablation of its receptor GC-B results in an increased proportion of

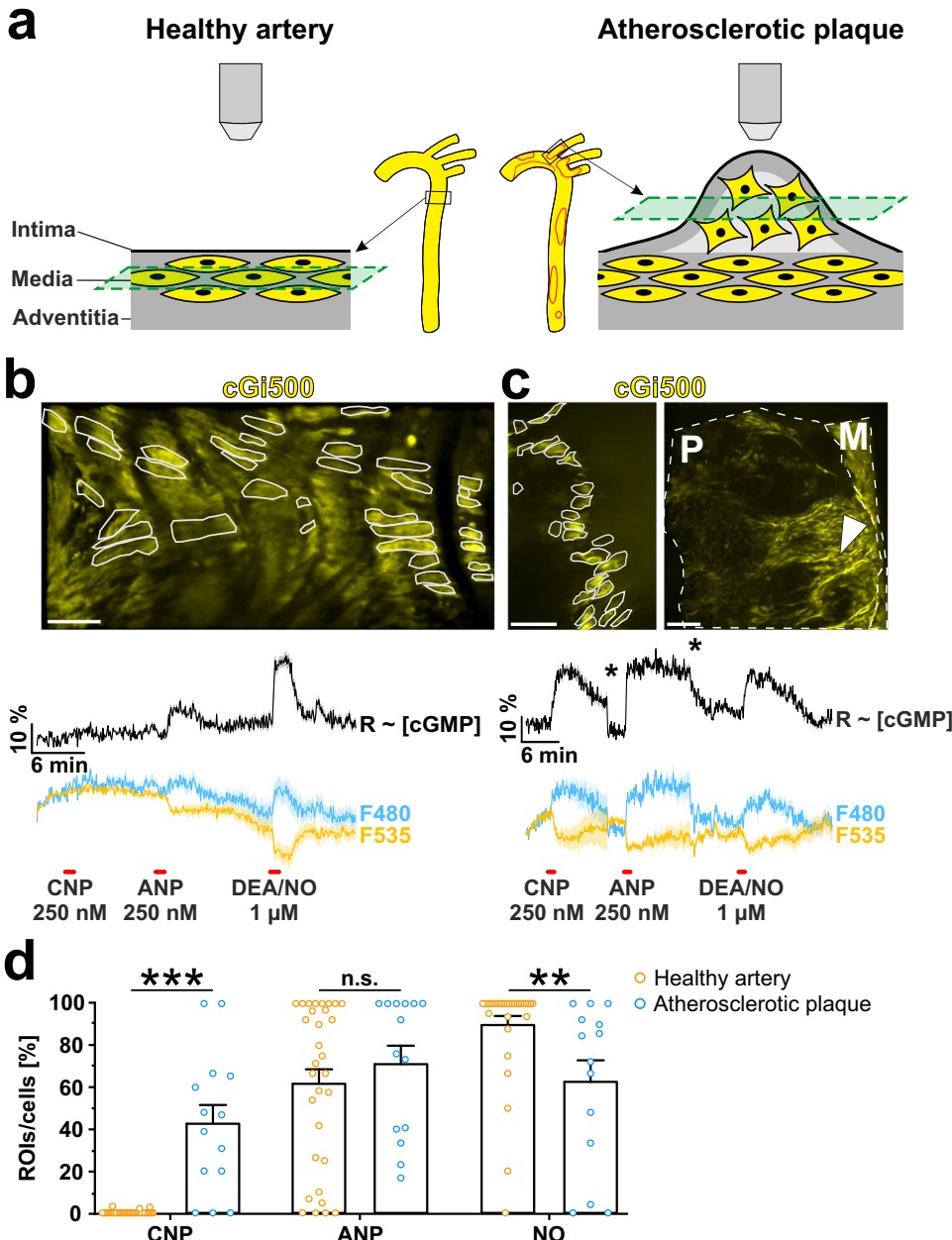

**Fig. 5 | cGMP imaging in healthy and atherosclerotic arteries from global cGMP sensor mice. a** Schematic representation of a healthy (left) and an atherosclerotic (right) aorta expressing the cGi500 sensor (yellow) as used for ex vivo cGMP/FRET measurements. Atherosclerotic plaques are outlined in red. The image planes are indicated in green. Arteries were measured from the luminal side as indicated by the objective. Representative cGMP/FRET measurements of (**b**) a healthy and (**c**) an atherosclerotic artery. Tissues were successively stimulated with CNP, ANP, and DEA/NO (red bars, concentrations indicated in the panel). The black traces indicate the intracellular cGMP concentration over time (ratio trace of CFP/YFP or R ~ [cGMP]); cyan and yellow traces show CFP and YFP fluorescence of the sensor, respectively. Shown are means ± SEM (shadow behind the trace) of representative response patterns (healthy artery: $n = 14$ regions; atherosclerotic plaque: $n = 7$ regions). The black scale bars indicate the time and percent change of the traces relative to baseline. Asterisks indicate refocus events. The images show (**b**) a healthy aorta and (**c**, left) an atherosclerotic plaque during a cGMP/FRET measurement, and (**c**, right) a maximum intensity projection of the measured

atherosclerotic plaque. Sensor-expressing cells are visualized by the YFP fluorescence of cGi500. ROIs/cells used for the analysis of cGMP responses are outlined in gray. White scale bars from left to right are 20 μm, 50 μm, and 100 μm. Dashed lines outline plaque (P) and media (M). The white arrowhead indicates the region that was measured. **d** Percentage of ROIs/cells per measurement that showed a cGMP response to CNP, ANP, and/or NO in healthy (orange) and atherosclerotic (cyan) arteries. Data are shown as mean + SEM (healthy arteries: 524 ROIs/cells were analyzed out of 715 recorded ROIs/cells in 31 measurements from 24 healthy aortae; atherosclerotic plaques: 193 ROIs/cells were analyzed out of 337 recorded ROIs/cells in 14 measurements from six atherosclerotic aortae). Statistical significance is indicated by asterisks (**$p < 0.01$; ***$p < 0.001$; n.s.: not significant). As detailed in the Methods section, ROIs/cells showing a poor quality of their single and/or ratio traces were excluded from analysis. The respective numbers of analyzed ROIs/cells out of total recorded ROIs/cells are indicated above. Source data including exact p-values and applied statistical tests are provided in the Source Data file.

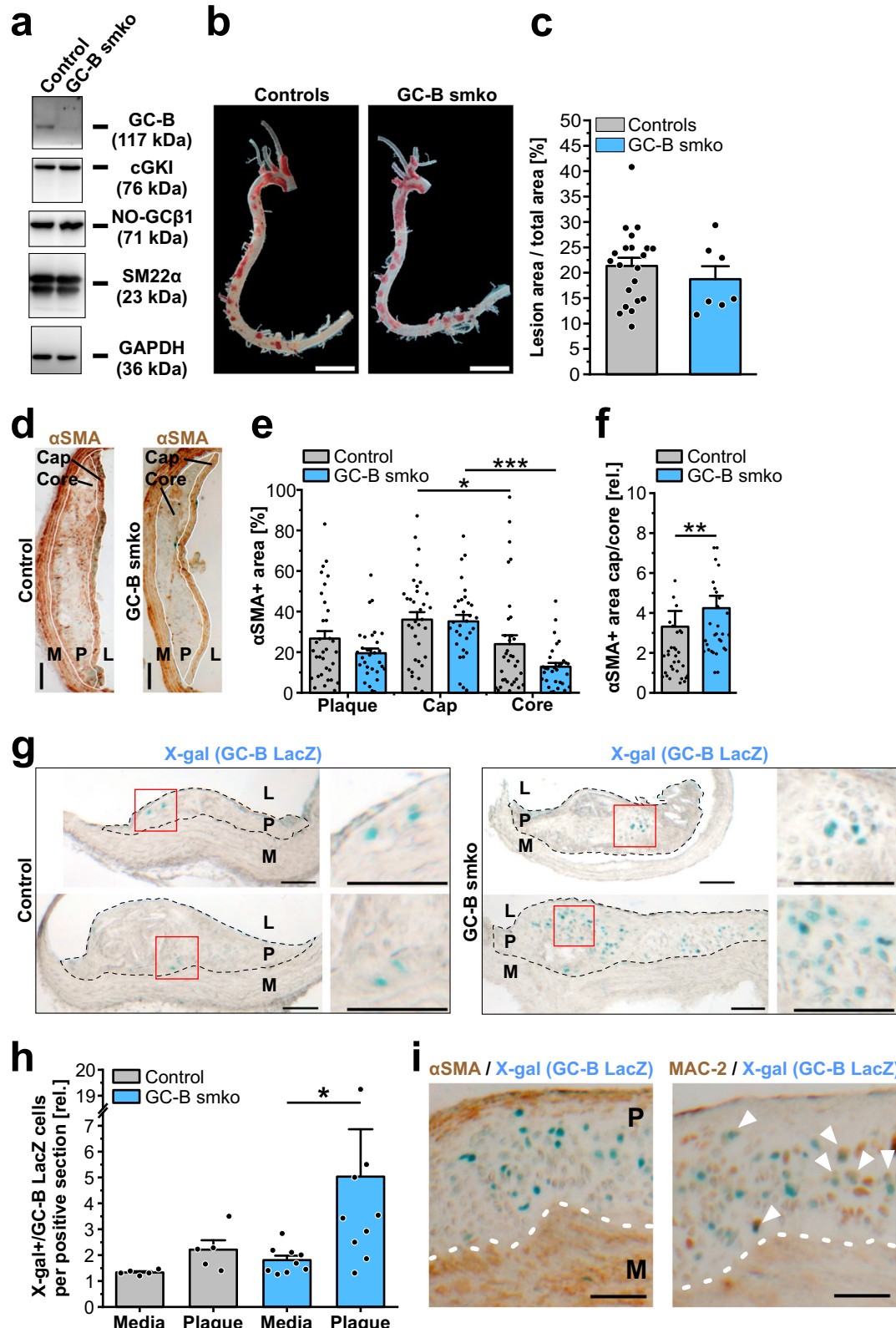

modulated VSMCs in the plaque core region. Indeed, X-gal⁺ cells in the plaque core frequently lacked expression of αSMA but expressed MAC-2 (also known as Lgals3) (Fig. 6i), a marker of modulated VSMCs with macrophage-like[69] and chondromyocyte[65] phenotypes.

Consistent with the similar lesion area in control and GC-B^smko mice (Fig. 6c), the number of cell nuclei per plaque section did not differ significantly between genotypes (Fig. 7a). This indicated that the

GC-B knockout in VSMCs did not have a strong effect on plaque cell proliferation and/or migration. However, ablation of GC-B in VSMCs promoted a more fibrotic/osteochondrogenic plaque phenotype. 42% ± 5% of X-gal⁺ cells in plaque sections were positive for the chondrocyte marker osteopontin (Fig. 7b). Importantly, significantly more osteopontin⁺ cells as well as increased collagen deposition were detected in X-gal⁺ plaques of GC-B^smko mice as compared to control

**Fig. 6 | Lesion area, plaque composition and GC-B expression in atherosclerotic aortae of control and GC-B smko mice after 18 weeks of atherogenic diet.**
**a** Analysis of GC-B ablation by Western blotting. Applied antibodies and expected molecular weight of the target proteins are indicated at the right. Each lysate was obtained from two pooled aortae. No further biological replicates were performed. **b**, **c** Effect of SMC-specific GC-B ablation on lesion area of atherosclerotic aortae. **b** Representative images of atherosclerotic aortae after Oil Red O staining. Atherosclerotic lesions appear red. Scale bars, 4 mm. **c** Relative lesion area in percent of total vessel area. Each data point represents an aorta. Data are shown as mean + SEM (controls: 21 aortae; GC-B smko: 7 aortae). **d**–**f** Effect of SMC-specific GC-B ablation on distribution of αSMA+ cells in atherosclerotic plaques. **d** Representative images of immunohistochemical stainings of plaque sections for αSMA (brown). The 30-μm thick cap and the core area (equivalent to the non-cap region of the plaque) are outlined by a white line. Scale bars, 100 μm. **e** Quantification of αSMA+ area in different plaque regions. **f** Cap-to-core distribution of αSMA in atherosclerotic lesions. Each data point represents an individual plaque. Data is shown as mean + SEM (control: 34 plaques from six mice; GC-B smko: 31 plaques from six mice). Datapoints above 10 are not shown (control: 3, GC-B smko: 2). Statistical significance is indicated by asterisks (*$p < 0.05$; **$p < 0.01$; ***$p < 0.001$). **g**, **h** Effect of SMC-specific GC-B ablation on the number of X-gal + / GC-B LacZ cells expressing the GC-B LacZ reporter. Note that the X-gal + /GC-B LacZ

cells in plaques likely represent modulated VSMCs. **g** Representative images of sections from atherosclerotic plaques after X-gal staining. Two images per genotype are shown to demonstrate the inter-plaque heterogeneity of GC-B expression. X-gal+ cell nuclei appear as cyan dots. Red boxes indicate the plaque region that is magnified at the right. Scale bars, 100 μm. **h** Quantification of X-gal + /GC-B LacZ cells in the media and plaque region. Each data point is based on all sections prepared from one aortic arch. Data are shown as mean + SEM (control: five aortic arches; GC-B smko: nine aortic arches). Statistical significance is indicated by asterisks (*$p < 0.05$). Note that the power for the comparison within the control group was below 80%. **i** Expression of αSMA and MAC-2 in X-gal+ GC-B LacZ reporter cells found in atherosclerotic plaques. Images of immunohistochemical stainings (brown) for αSMA (left) and MAC-2 (right) were performed on X-gal-stained plaque sections obtained from GC-B smko mice that also carried the GC-B LacZ reporter gene. The border between plaque and media is indicated by a dashed white line. X-gal+ cell nuclei appear as cyan dots. White arrowheads indicate MAC-2/X-gal double positive cells (right). Note that there were no αSMA/X-gal double positive cells (left). Results are representative for 123 sections from 8 mice. Data points shown in (**c**, **e**, **f**, **h**) are pooled from male and female mice. Scale bars, 50 μm. M, media; P, plaque; L, lumen of the vessel. Source data including exact p-values and applied statistical tests are provided in the Source Data file.

mice (Fig. 7c–e). Moreover, calcification of X-gal$^+$ lesions was rarely detected in control mice, while we frequently found calcified plaques in aortic arch sections from GC-B$^{smko}$ mice (Fig. 7f; g). Together, our in vivo data indicated that genetic ablation of GC-B in VSMCs promotes their transdifferentiation to bone-like plaque cells, also called "chondrocyte-like cells" or "fibrochondrocytes", without changing the absolute number of lesional cells.

## Discussion

The present study highlights fluorescent cGMP biosensors as powerful tools for the sensitive analysis of cGMP signaling pathways in individual cells and supports the notion that single-cell analysis can lead to new concepts in vascular biology[73]. Using cGMP imaging in real time in living VSMCs combined with immunostaining and functional analyses, we found a marked heterogeneity of ANP- and CNP-induced pathways in VSMCs and identified the CNP/GC-B/cGMP axis as a marker and regulator of modulated VSMCs in atherosclerosis.

The cyclic nucleotide cGMP is critical for the maintenance of cardiovascular health, and various components of the cGMP system are established clinical targets in coronary artery disease, pulmonary hypertension, or heart failure. However, it is not completely understood which cell types and pathways contribute to cGMP's beneficial effects on the cardiovascular system. VSMCs are long known as important hubs of cGMP signaling. It is assumed that NO-GC, GC-A and GC-B are co-expressed in VSMCs and that NO, ANP and CNP may regulate different cell functions via different targets[31,53]. But how can different guanylyl cyclase ligands exert different functional effects if they all increase the intracellular cGMP level? If we hypothesize, for instance, that ANP and CNP exert different effects in the same cell, this may be explained by compartmentalization of cGMP signaling complexes targeted by ANP vs. CNP stimulation in this individual cell, or the effects of ANP or CNP may be mediated by mechanisms that are at least partially independent of cGMP. If we assume that GC-A and GC-B are differentially expressed in different cells of the investigated cell population, as shown in the present study, then the differential effects induced by ANP vs. CNP could be related to a variability of the respective downstream signaling pathways between individual cells of the investigated cell population or tissue. Thus, a major question in cGMP research is whether differential functional responses to cGMP-elevating ligands are due to heterogeneity of downstream signaling *within* the same cell (compartmentalization) or *between* individual cells of the analyzed population (cell-to-cell variability)[53]. Based on our single-cell analyses (Figs. 1–4), we propose that at least one explanation

for differential effects of ANP and CNP on VSMCs is the heterogeneity of cGMP signaling pathways *between* individual VSMCs with different phenotypic states.

We used live-cell cGMP imaging as a highly sensitive method to detect the enzymatic activity of NO-GC, GC-A, and GC-B in individual VSMCs. We found a marked cell-to-cell variability of ANP- vs. CNP-induced cGMP pathways in VSMC populations grown in vitro. Correlative single-cell profiling of cGMP signals and VSMC phenotype by cGMP imaging and immunofluorescence staining of marker proteins demonstrated that cGMP responses are strongly associated with the cell's phenotypic state. The ANP/GC-A axis is predominantly expressed in contractile/differentiated VSMCs, whilst the presence of a functional CNP/GC-B cascade is a characteristic feature of modulated VSMCs. The upregulation of the CNP/GC-B/cGMP pathway in modulated VSMCs in vitro was confirmed by complementary single-cell analyses using scRNA-seq and a LacZ reporter inserted into the GC-B gene locus. Importantly, monitoring of GC-B activity in atherosclerotic plaques using real-time cGMP imaging and the genetic LacZ reporter indicated that phenotypic modulation of VSMCs that occurs during the progression of atherosclerosis leads to the development of GC-B positive plaque cells in vivo (Figs. 5, 6). Due to the technical limitations of optical imaging described in the Results section, we were unable to characterize cGMP signaling in cells of the inner plaque core by cGMP/ FRET imaging. However, by using the genetic lacZ reporter as an alternative method to detect GC-B-expressing cells, we were able to clearly demonstrate that cells with high GC-B expression also exist in the inner plaque core (Fig. 6g, i).

Using cGMP imaging, we did not observe a similar upregulation of the CNP/GC-B system in a mouse model of restenosis, in which vascular remodeling and formation of a neointima is induced by injury of the carotid artery. While both healthy and injured arteries showed cGMP increases in response to ANP and NO, neither reacted to CNP (Supplementary Fig. 8). These results suggest that the role of CNP-induced cGMP in vascular remodeling is context-specific, being more important in atherosclerosis than in restenosis. This view is also supported by previous studies showing that SMC-specific deletion of the cGMP downstream effector cGKI in mice altered the development of atherosclerosis[74] but not restenosis[75].

The findings of the present study are consistent with a previous report describing downregulation of GC-A and upregulation of GC-B expression in smooth muscle cells of human atherosclerotic lesions[76]. Our analyses of atherosclerotic lesions are also in agreement with recent RNA sequencing studies. Mokry et al. reported GC-B expression

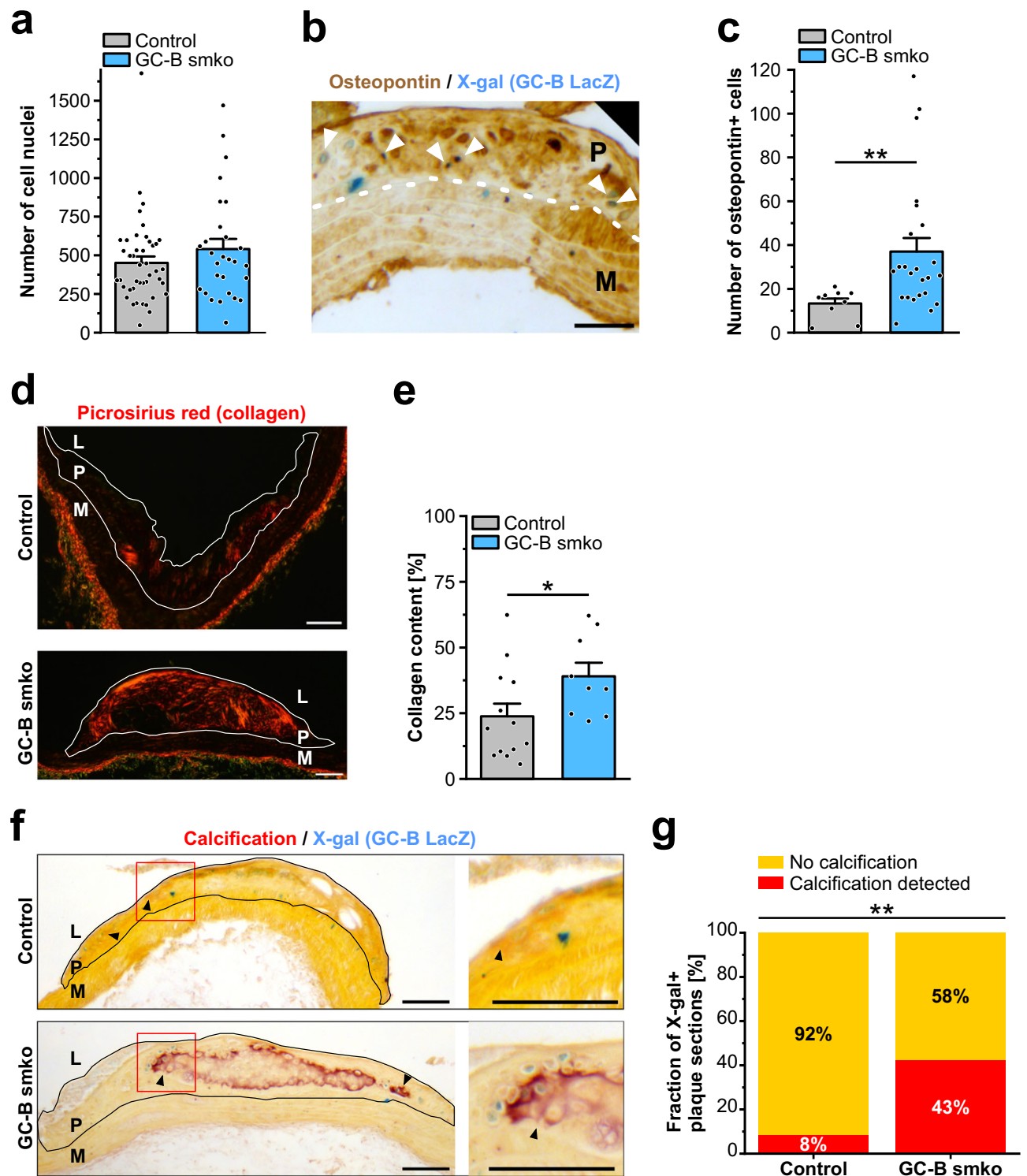

in so-called fibro-inflammatory human plaques[77], and Pan et al. and Alencar et al. detected GC-B mRNA in mouse and human plaques, specifically in populations of modulated VSMCs classified as fibrochondrocytes[78] or chondrocyte-like cells[79]. In agreement with the findings of the present study, our re-analysis of their scRNA-seq datasets (Supplementary Fig. 9) showed that GC-B (*Npr2*) mRNA is enriched in the murine fibrochondrocyte cluster, while GC-A (*Npr1*) is strongly expressed in contractile SMCs and downregulated in fibrochondrocytes. In the human dataset, GC-B mRNA expression appeared to be relatively low but detectable, and the distribution of GC-B vs. GC-A

mRNA was similar to murine lesions (Supplementary Fig. 9). Thus, the CNP/GC-B axis appears to be upregulated in VSMC-derived plaque cells with bone cell-like properties. This indicates similarities between the CNP/GC-B systems in VSMC-derived plaque cells and bona fide bone cells. Future studies should investigate whether CNP controls vascular and bone remodeling via similar or different molecular mechanisms and whether our findings in mice are also relevant for atherosclerosis in humans.

The present study has shown that cGMP in contractile VSMCs is mainly generated by the ANP- and NO-dependent pathways, while

**Fig. 7 | Cell numbers, chondrongenic, and fibrotic parameters of plaques from atherosclerotic aortae of control and GC-B smko mice after 18 weeks of atherogenic diet. a** Effect of SMC-specific GC-B ablation on cell number as indicated by the number of nuclei (stained with Hoechst No. 33258) per plaque section. Each data point represents an individual plaque. Data is shown as mean + SEM (control: 43 plaques from five mice; GC-B smko: 28 plaques from six mice). **b** Expression of osteopontin in X-gal+ GC-B LacZ reporter cells found in atherosclerotic plaques. Immunohistochemical stainings for osteopontin (brown) were performed on X-gal-stained plaque sections from GC-B smko mice that also carried the GC-B LacZ reporter gene. The border between plaque and media is indicated by a dashed white line. X-gal+ cell nuclei appear as cyan dots. White arrowheads indicate osteopontin/X-gal double positive cells. Scale bar, 50 μm. **c** Effect of SMC-specific GC-B ablation on the number of osteopontin+ cells in X-gal+ plaque sections. Each data point represents an individual plaque section. Data is shown as mean + SEM (control: 9 plaque sections from three mice; GC-B smko: 24 plaque sections from seven mice). Statistical significance is indicated by asterisks (**p < 0.01). **d, e** Effect of SMC-specific GC-B ablation on deposition of collagens in atherosclerotic plaques. **d** Representative images of picrosirius red stainings of X-gal+ plaque sections for collagens (red) under cross-polarized light. Plaques are outlined by a white line. Scale bars, 100 μm. **e** Quantification of collagen+ area. Each data point represents an individual plaque. Data is shown as mean + SEM (control: 13 plaques from five mice; GC-B smko: nine plaques from four mice). Statistical significance is indicated by an asterisk (*p < 0.05). **f, g** Effect of SMC-specific GC-B ablation on plaque calcification. **f** Representative images of Alizarin Red S staining of X-gal+ plaque sections for calcium (red). Plaques are outlined by a black line. X-gal+ cell nuclei appear as cyan dots. Black arrowheads indicate calcified plaque areas, which are hardly visible in lesions from control mice. Red boxes indicate the plaque region that is magnified at the right. Scale bars, 100 μm. **g** Quantification of plaque sections that were positively stained for calcium (red) or showed no signs of calcification (yellow) (control: 24 plaque sections from four mice; GC-B smko: 80 plaque sections from nine mice). Statistically significant differences in the distribution of calcified and non-calcified plaque sections between control and GC-B smko mice are indicated by asterisks (**p < 0.01). Data shown in panels (**a, c, e, g**) are pooled from male and female mice. M, media; P, plaque; L, lumen of the vessel. Source data including exact p-values and applied statistical tests are provided in the Source Data file.

phenotypic modulation in cell culture and in the context of atherosclerosis switches VSMCs to CNP-dependent cGMP signaling. This is also consistent with previous studies that performed bulk analysis of GC-A and GC-B mRNA expression in rat aortic SMCs. These studies showed that passaging of VSMCs, presumably promoting phenotypic modulation of the cells, was associated with a decrease of GC-A mRNA and an increase of GC-B mRNA[80], and that the relative mRNA level of GC-B in passaged VSMCs was much higher than GC-A[81]. Thus, we propose GC-B as a marker for modulated VSMCs in vitro and in vivo. In the present study, we detected GC-B at the single-cell level by measuring its gene activity using a LacZ reporter, its RNA level using scRNA-seq or, with high sensitivity, its enzymatic activity using cGMP/FRET imaging in living cells and tissues. For routine detection of GC-B as a marker, the use of the LacZ reporter system, scRNA-seq, or FRET microscopy is certainly too laborious. Available antibodies against GC-B were suitable for Western blot analysis, which can already provide an estimate of the presence of modulated VSMCs in a sample (see, e.g., Fig. 4e). Future efforts are needed to develop antibodies that allow routine detection of GC-B at the cellular level as a marker for modulated VSMCs.

To study the functional relevance of the CNP/GC-B/cGMP pathway in VSMCs, we generated SMC-specific GC-B knockout mice. The analysis of atherosclerosis in these mouse mutants revealed that intact CNP/GC-B signaling in VSMCs might be atheroprotective by attenuating the development of potentially harmful VSMC-derived plaque cells. Recent studies using cell lineage tracking discovered a pronounced plasticity of VSMCs in atherosclerotic lesions in vivo and suggested that these cells can switch via a multipotent transitional state to a number of cell phenotypes including macrophage-like, fibroblast-like, and osteoblast-like plaque cells[82–84]. In the present study, we found that plaques from mice lacking GC-B in VSMCs had an increased proportion of modulated VSMCs with bone cell-like properties, leading to the development of potentially unstable plaques with an osteochondrogenic phenotype (Figs. 6, 7). The GC-B knockout phenotype indicates that activation of the CNP/GC-B/cGMP axis normally inhibits the formation and/or expansion of chondrocyte-like plaque cells. In line with these results, others reported that CNP inhibits vascular calcification by attenuating pathological remodeling of aortic valve interstitial cells to myofibroblasts and osteoblast-like cells[85] and by negatively regulating the osteogenic transition of VSMCs[86] through the GC-B/cGMP pathway. Moreover, the incidence of aortic valve disease, which shares risk factors with atherosclerosis, was significantly higher in male GC-B[+/-] mice than in wildtype mice[87]. A recent study reported that CNP administration via osmotic pumps or CNP overexpression attenuated atherosclerosis and increased plaque stability in ApoE-deficient mice[88].

Together, the findings by us and others reveal an important protective role of the CNP/GC-B/cGMP axis as a regulator of cell phenotypes that arise in the context of vascular disease. Given the known role of the CNP/GC-B/cGMP axis in bone growth[11,18], it is perhaps not surprising that this signaling cascade is also expressed in chondrocyte-like cells of the diseased vasculature and that CNP regulation of these cells influences plaque stability and vascular calcification. However, the molecular mechanisms of CNP action appear to differ between arteries and bones. While CNP activates ossification and calcification in bone, it seems to inhibit these processes in the vascular wall.

The in vitro and in vivo findings of this study support the concept of heterogeneity of ANP- and CNP-induced cGMP signaling in VSMCs with different phenotypic states. In our current model (Fig. 8), the ANP/GC-A axis predominates in contractile VSMCs and likely mediates ANP-induced vasodilation and acute blood pressure changes[27], while the CNP/GC-B axis is strongly upregulated in modulated bone-like VSMCs, where it might mediate the reported anti-proliferative and anti-migratory effects of CNP (Fig. 8a)[41,43,44]. In the setting of atherosclerosis, the CNP/GC-B/cGMP axis is functionally present in the plaques. CNP is released from lesional ECs and VSMCs[76] and acts locally on the VSMC-derived transitional/modulated VSMCs expressing GC-B. At least a fraction of the GC-B positive plaque cells has a chondrocyte-like phenotype. The resulting increase of intracellular cGMP then attenuates the expansion of these cells to potentially harmful plaque cells (Fig. 8b). This effect of CNP should counteract the destabilization of plaques and therefore be atheroprotective.

In the future, it will be of interest to study the effects of CNP and GC-B on the in vivo behavior of modulated plaque VSMCs in detail. For example, effects on clonal expansion of VSMC-derived cells could be investigated using a recently established PET reporter system for non-invasive monitoring of clonal cell patches in mice[89]. We also need to delineate the downstream mechanisms of CNP/GC-B/cGMP signaling in modulated VSMCs. The effects of cGMP may well be mediated by its canonical receptor in VSMCs, cGKI. Indeed, this protein kinase has already been implicated in the regulation of VSMC growth and phenotype in atherosclerotic plaques[74]. Degradation of cGMP by PDEs is important to adjust intracellular cGMP concentrations in the GC-B positive VSMCs. According to our scRNA-seq data, these PDEs could be Pde1a, Pde3a and Pde5a. In addition, Pde10a might be involved in antagonizing the effects of the CNP/GC-B/cGMP/cGKI pathway on VSMC growth and migration[81], even though we could only detect low levels of Pde10a by scRNA-seq in our VSMCs (Fig. 3d). Our results do not exclude the possibility that CNP, in addition to its cGMP-dependent effects via activation of GC-B, might also exert cGMP-independent actions on vascular remodeling via binding to Npr-C[37,38,45]. Indeed, Npr-C (*Npr3*) mRNA was highly expressed in our

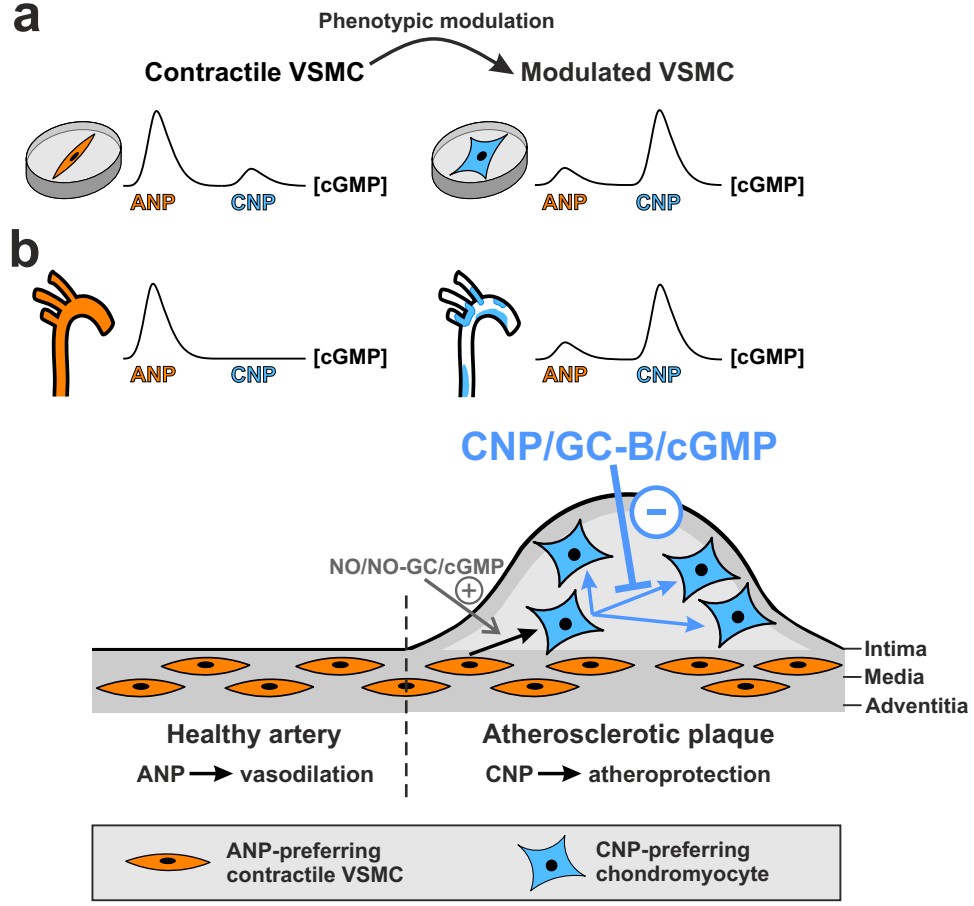

**Fig. 8 | Graphical summary of the major findings of the study.** These findings led to the concept of heterogeneity of ANP- and CNP-induced cGMP signaling in VSMCs with different phenotypic states and an atheroprotective role of the CNP/GC-B/cGMP axis in modulated VSMCs. Using correlative single-cell cGMP imaging and immunostaining of marker proteins, we found that the capacity to generate cGMP via the ANP/GC-A and CNP/GC-B axis changes dramatically with the phenotypic state of VSMCs. **a** In culture, contractile VSMCs (orange) are more sensitive to ANP than to CNP, while modulated VSMCs (cyan) respond much better to CNP than to ANP. **b** As predicted by the cell culture model, pathological remodeling from healthy to atherosclerotic arteries shifts ANP-dependent to CNP-dependent cGMP signaling, which we propose as a marker for modulated VSMCs in vitro and in vivo. SMC-specific ablation of GC-B suggests that an active CNP/GC-B/cGMP signaling axis in modulated VSMCs, including chondromyocytes, prevents their further proliferation and transition to potentially deleterious lesional cells (cyan inhibitory arrow). This effect of CNP should counteract destabilization of plaques and should therefore be atheroprotective. Previous studies have shown that ANP induces vasodilation and that activation of the NO/NO-GC/cGMP axis in VSMCs promotes the growth of VSMC-derived plaque cells (gray stimulatory arrow).

modulated VSMCs (Fig. 3d), and expression of Npr-C seems to be strongly upregulated in parallel with GC-B in VSMCs of human coronary atherosclerotic lesions[50,76]. However, genetic ablation of Npr-C in mice decreased atherosclerotic lesion burden[47], whilst EC-specific deletion of CNP increased it[38]. These data argue against the view that EC-derived CNP is atheroprotective through a mechanism involving Npr-C.

Collectively, our study identifies the CNP/GC-B/cGMP signaling axis as a marker of phenotypically modulated VSMCs that develop in vitro during cell culture and in vivo during formation of atherosclerotic lesions, particularly chondrocyte-like plaque cells. Functionally, it provides a mechanism to limit the extent of VSMC phenotypic modulation during atherogenesis (Fig. 8b). The atheroprotective role of the CNP/GC-B system in modulated VSMCs adds a further building block to our understanding of cGMP signaling in atherosclerosis. Other studies have indicated that NO, via the NO-GC/cGMP/cGKI pathway in VSMCs, can enhance VSMC growth and phenotypic modulation in atherosclerotic plaques[74,90] (Fig. 8b). On the other hand, activation of the NO-GC/cGMP axis in platelets can attenuate atherosclerosis via paracrine mechanisms[91]. Thus, a picture emerges in which cGMP can positively and negatively influence atherosclerosis via different cell types and signaling pathways, further supporting the concept of cGMP signaling heterogeneity in the cardiovascular system[53] and beyond[92]. In the future, we should further investigate the similarities and differences between cGMP signaling in bone and vascular remodeling. It would be particularly interesting to test whether the activity of the CNP/GC-B/cGMP pathway in atherosclerotic arteries can be pharmacologically enhanced with GC-B agonists such as vosoritide, which is used to treat bone disease, and whether this could attenuate atherosclerosis.

## Methods
### Experimental animals
All animal experiments were approved by the local authority (Regierungspräsidium Tübingen). Mice were housed with *ad libitum* access to food and water with a 12 h light/dark cycle. For cGMP/FRET measurements, mice were used carrying either a globally expressed transgene encoding the cGMP biosensor cGi500 (cGi500-L1 mice) or a loxP-flanked sensor transgene that can be expressed in a cell type-specific manner after Cre-dependent recombination (cGi500-L2 mice)[54]. cGi500-L2 mice were crossed to SM22-Cre mice[27] to generate SMC-specific cGMP sensor mice (genotype: SM22-Cre[+/tg]; cGi500[+/L2]) used

for cGMP imaging in cell culture and healthy aortae. For SMC-specific cGi500 expression in the atherosclerotic aorta, cGi500-L2 mice were crossed to SMA-CreER[T2] mice[68] and ApoE-deficient mice[93] (genotype: ApoE[-/-]; SMA-CreER[T2 +/tg]; cGi500[+/L2]). Sensor expression was induced in these mice by tamoxifen injection (5 × 1 mg/mouse/day at 4 weeks of age). To detect GC-B-expressing cells, GC-B LacZ reporter mice expressing a nuclear targeted β-galactosidase under control of the GC-B promoter (GC-B[+/LacZ]) were used[66]. Note that the LacZ reporter gene was integrated into the endogenous GC-B locus, so that GC-B expression from the LacZ reporter allele was dysfunctional. Homozygous GC-B LacZ mice also expressing the cGMP sensor (GC-B[LacZ/LacZ]; cGi500[+/L1]) were used as GC-B knockout model to investigate cGMP responses in primary VSMCs. To analyze the role of GC-B in VSMCs in atherosclerosis, mice carrying a loxP-flanked GC-B allele (GC-B[+/L2])[94], the GC-B LacZ reporter (GC-B[+/LacZ]), and the SMA-CreER[T2] recombinase were bred on an ApoE-deficient background. This resulted in pro-mutant SMC-specific GC-B knockout mice (genotype: ApoE[-/-]; GC-B[L2/LacZ]; αSMA-CreER[T2 +/tg]) and control mice (genotype: ApoE[-/-]; GC-B[L2/LacZ]; αSMA-CreER[T2 +/+] or ApoE[-/-]; GC-B[L2/+]; αSMA-CreER[T2 +/tg] or ApoE[-/-]; GC-B[L2/+]; αSMA-CreER[T2+/+]). SMC-specific ablation of GC-B was achieved by Cre-mediated conversion of the loxP-flanked GC-B allele into a nonfunctional GC-B knockout allele, selectively in SMCs. To trigger the conversion, mice were injected intraperitoneally with tamoxifen (5 × 1 mg/mouse/day at 4 weeks of age). Note that control mice were also injected with tamoxifen to control for potential side effects of tamoxifen not related to the GC-B knockout.

Male and female mice were used for experiments. All atherosclerotic mice had a mixed 129 Sv/C57BL6/N background. All other mice had a pure C57BL6/N background.

For cGMP/FRET measurements, transgenic mice and cells expressing the cGMP sensor were used. For cGMP imaging in cell culture, VSMCs were isolated from mice aged between 3 and 17 weeks. For cGMP/FRET measurements in healthy aortae, mice between 7 and 32 weeks were used, and for cGMP/FRET imaging of atherosclerotic arteries, mice between 38 and 61 weeks were used. To analyze the impact of GC-B-dependent cGMP signaling in VSMCs on athero-sclerosis, 26-week-old mice were used. As a mouse model of restenosis, carotid artery ligation was performed as described[75,95]. Briefly, the left common carotid artery of deeply anesthetized 12-week-old mice was dissected and thoroughly freed of fat and surrounding tissue using atraumatic forceps. The blood flow was then obstructed by a ligature placed near the bifurcation. After closing the wound, the mice received carprofen (10 mg/kg) and were transferred to their home cages for recovery with free access to food and water. The animals showed no signs of stroke or other abnormalities during the entire follow-up period. cGMP/FRET measurements in restenotic blood vessels were performed 4 weeks after carotid injury.

## Reagents and antibodies
Bovine serum albumin (BSA; #8076.4) and phenylmethylsulfonyl-fluoride (PMSF, #6367.1) were purchased from Roth; atrial natriuretic peptide (ANP; 1-28, #1912) and C-type natriuretic peptide (CNP; 1-22, #3520) from Tocris; diethylammonium (Z)-1-(N,N-diethylamino)dia-zen-1-ium-1,2-diolate (DEA/NO, #ALX-430-034-M010) from Axxora; Dulbecco's modified Eagle medium (DMEM; with Glutamax™-I, 4.5 g/L D-glucose, sodium pyruvate, phenol red, #31966-021), fetal calf serum (FCS, #10270106, Lot-41G6601K), 100x Pen/Strep (10,000 U/mL penicillin, 10,000 µg/mL streptomycin, #15140122) and trypsin (#15400054) from Gibco; hyaluronidase from bovine testes (#H3506), collagenase from clostridium histolyticum (#C7926), papain from papaya latex (#P4762), Hoechst No. 33258 (#14530), 3,3′-diamino-benzidene (DAB, #D5637), Oil Red O (#O0625), tamoxifen (#T5648-1G), saturated picric acid solution (#P6744), and Direct Red 80 (#365548) from Sigma-Aldrich; Alizarin Red S (#042040.14) from Thermo Scientific; normal donkey serum (NDS; #S30-100ML) and normal goat serum (NGS; #S26-100ML) from Millipore; 5-bromo-4-chloro-3-indolyl-β-D-galactopyranoside (X-gal, #15520018) from Invitrogen; Surgipath Paraplast X-tra (Paraffin, #39603002) from Leica; standard chow (#1324) and atherogenic diet (modified #1324: 20% fat, 1.5% cholesterol) from Altromin.

Antibodies directed against the following proteins were used for Western blotting (WB), immunofluorescence (IF), and immunohis-tochemistry (IHC): SM22α (1:1000 WB, 1:500 IF, rabbit, #ab14106) and αSMA (1:2000 IHC, rabbit, #ab124964) from Abcam; MAC-2 (1:200 IHC, rat, #Cl8942) from Cedarlane; PDGFRα (1:500 IF, rabbit, #3174), GAPDH (1:1000 WB, rabbit, #2118) and horse radish peroxidase (HRP)-conjugated anti-rabbit antibody (1:5000 WB, goat, #70749) from Cell Signaling; HRP-conjugated anti-guinea pig antibody (1:10,000 WB, donkey, #706-035-148) from Dianova; Alexa Fluor 488-conjugated anti-mouse antibody (1:500 IF, goat, #A11029), Alexa Fluor 555-conjugated anti-mouse antibody (1:500 IF, goat, #A21424), Alexa Fluor 488-conjugated anti-rabbit antibody (1:500 IF, goat, #A11008), and Alexa Fluor 555-conjugated anti-rabbit antibody (1:500 IF, goat, #A21428) from Life Technologies; S100A4 (1:300 IF, rabbit, #07-2274) from Millipore; osteopontin (1:100 IHC, goat, #AF808) from R&D Systems; αSMA (1:500 IF, mouse, #A2547) from Sigma-Aldrich; bioti-nylated anti-goat antibody (1:250, horse, #BA-9500), biotinylated anti-rabbit antibody (1:250 IHC, goat, #BA-1000), and biotinylated anti-rat antibody (1:250 IHC, rabbit, #BA-4001) from Vector Laboratories. The GC-B antibody (1:5000 WB, guinea pig) was from Hannes Schmidt[66]; the NO-GCβ1 1 A antibody (1:800 IF, rabbit) and NO-GCβ1 2 A antibody (1:10,000 WB, rabbit) was a gift from Andreas Friebe[96]. The cGKI antibody (1:5000 WB, rabbit) was generated in the Feil laboratory[97].

## X-gal staining
To detect LacZ activity by X-gal staining, cultured VSMCs or aortae were fixed with cell fix (2% w/v formalin, 0.2% w/v glutaraldehyde in phosphate-buffered saline [PBS]) for 5 min and 3 min, respectively, washed with PBS, and incubated with X-gal staining solution (1 mg/mL X-gal, 2 mM MgCl$_2$, 2.5 mM K$_3$Fe(CN)$_6$, 2.5 mM K$_4$Fe(CN)$_6$ in PBS) overnight at 37 °C. Staining was stopped by washing with PBS. Then, cultured VSMCs were incubated for 30 min with Hoechst No. 33258 (1 µg/mL) in BSA-PBS (0.5% w/v BSA in PBS), mounted in 80% (v/v) glycerol, and coverslips were sealed with transparent nail polish. Aortae were postfixed with cell fix for 10 min, washed with PBS, and further processed as described under "Analysis of atherosclerosis".

X-gal stainings were documented with an Axioskop 20 (ZEISS), equipped with an A-Plan 5x/0.12, an A-Plan 10x/0.5, and a Plan NeoFluar 20x/0.5 objective (ZEISS). For bright field illumination, a tungsten halogen lamp was used. For fluorescence illumination of Hoechst No. 33258 nuclear counterstaining, an HBO 50 mercury-vapor short-arc lamp was used with SP 365/LP 420 excitation/emission filter and a DCLP 395 dichroic mirror. Sections were documented with an EOS 750 D digital camera (Canon). Stained primary VSMCs were docu-mented with a Marlin F-046C color charge-coupled device (CCD) camera (Allied Vision).

## Preparation and immunofluorescence staining of VSMC cultures
Primary VSMCs were obtained from aortae of wildtype and trans-genic mice (on average three aortae per cell preparation) and cul-tured as described[98] with minor adaptations. In short, mice were euthanized with CO$_2$, the thoracic aorta was isolated, cleaned from the surrounding tissue, and then dissociated into individual cells by enzymatic digestion at 37 °C (first with 0.7 mg/mL papain for 45 min, then with 1 mg/mL collagenase and 1 mg/mL hyaluronidase for 12 min). Finally, cells were resuspended in full medium (DMEM sup-plemented with 10% v/v FCS and 1x Pen/Strep) and 50,000 cells/well of a 24-well plate (equipped with 12 mm glass coverslips, #631-1577, VWR) or 180,000 cells/well of a 6-well plate (equipped with 25 mm coverslips), were plated. Primary VSMCs were grown for 5–9 days

and passaged VSMCs for up to two weeks at 37°C and 6% CO$_2$. Approx. 90% of our cultured cells were immuno-positive for the SMC marker αSMA. For passaging experiments, 800,000 VSMCs were plated in a 10 cm cell culture dish (Corning) and grown until they reached >50% confluency. Then, cells were detached with trypsin/EDTA, dissociated by pipetting and split at a ratio of 1:4–2:1 onto 10 cm cell culture dishes for further passaging or onto 6-well plates, equipped with 25 mm coverslips for cGMP/FRET measurements. VSMCs were passaged up to passage 4. Subconfluent cultures (50–90% confluence) were serum-starved in FCS-free medium (DMEM supplemented with 1x Pen/Strep) for 12–24 h before performing experiments, if not stated otherwise.

For immunofluorescence staining, VSMCs were fixed with formalin (3.7% w/v formalin in PBS) for 10 min. Cells were permeabilized and unspecific binding sites were blocked with Perm/Block solution (5% v/v NDS/NGS, 0.1% v/v Triton X-100 in BSA-PBS) for 10 min followed by incubation with primary antibodies (in 0.1% v/v Triton-X-100 in BSA-PBS) overnight at 4 °C. Then, cells were washed with BSA-PBS and antibodies were detected with fluorescently labeled secondary antibodies (in 1 μg/mL Hoechst No. 33258 in BSA-PBS) for 1 h. In case of double staining, the procedure was repeated from the blocking step with adjusted primary antibody incubation (2 h at room temperature). Finally, cells were mounted with Immu-Mount (#9990402, Thermo Scientific). Immunofluorescence stainings were documented with an Axiovert 200 (ZEISS) using the VisiView® software for image acquisition. The microscope was equipped with a Plan NeoFluar 10x/0.30 objective (ZEISS) and an electron-multiplying CCD (EM-CCD) camera (Retiga R1). A computer-controlled Oligochrome (TILL Photonics) with a Xenon short-arc lamp (UXL-S150MO, Ushio) served as light source. The following filter sets (excitation – dichroic mirror – emission) were used: BP 387/11 – 410 DCLP – LP 440 (Hoechst No. 33258); BP 497/16 – 516 DCLP – BP 535/22 (YFP, Alexa Fluor 488); BP 543/22 – 565 DCLP – BP 610/75 (mTomato, Alexa Fluor 555).

## Analysis of atherosclerosis

To induce atherosclerosis in mice, mice on an ApoE-deficient genetic background were fed an atherogenic diet (20% fat, 1.5% cholesterol; Altromin). For cGMP/FRET measurements, mice were fed the atherogenic diet at the age of 25 weeks for 16 weeks before experiments were performed. SMC-specific GC-B knockout and control mice received the atherogenic diet after tamoxifen-induced deletion of GC-B at the age of 8 weeks for 18 weeks. Then, mice were deeply anaesthetized and euthanized by collection of blood and perfusion fixation with cell fix for 3 min. To obtain serum from blood samples, whole blood was left for 2 h at room temperature to coagulate, centrifuged at 3000 g for 3 min, and the supernatant was stored at −80 °C. Serum total-, HDL- and LDL-cholesterol, as well as triglycerides were determined on the Atellica Solution clinical chemistry analyzer (Siemens Healthineers). Aortae were isolated, roughly cleaned from perivascular tissue, and stained with X-gal as described under "X-gal staining". Next, on a shaker, aortae were washed in 78% v/v methanol for 5 min, stained with Oil Red O solution (0.4% w/v Oil Red O, 222 mM NaOH in 78% v/v methanol) for 90 min, and finally destained with 78% v/v methanol for 5 min. For quantification of lesion area, Oil Red O-stained aortae were carefully cleaned from surrounding tissue, placed in a Sylgard 184 (#1673921, Dow) coated petri dish filled with water, and flattened with a glass coverslip, which was weighted with 10-g weights at its opposite corners. Aortae were documented from both sides with an EOS 750 D digital camera (Canon) mounted with a 1.6x lens on a stereomicroscope (Stemi 2000 CS, ZEISS). The lesion area of each side was determined with Fiji[99], normalized to the corresponding vessel area, and the mean of both sides of the aorta was calculated and used for further analyses. Lesion area was determined for the whole aorta and separately for the aortic arch (segment from up to 1 mm in front of the brachiocephalic artery and up to 1 mm rear of the left subclavian artery

and branching vessels) and thoracic/abdominal aorta (whole aorta without aortic arch).

After documentation of the lesion area, aortae were dehydrated by an increasing ethanol concentration row, finishing with a 2 min incubation in toluene. Finally, aortic arches were embedded in paraffin and cut in 10-μm sections. For immunohistochemical staining, sections of the aortic arch were deparaffinized and rehydrated. Next, sections were covered with Tris-buffered saline-Tween (TBS-T; 150 mM NaCl, 10 mM Tris, 0.1% v/v Tween-20; pH 8.0) and documented before further processing (see, "X-gal staining" for a description of the microscope). To stain cell nuclei, sections were incubated for 15 min in Tris-buffered saline-Tween supplemented with Hoechst No. 33258 (1 μg/mL) before documentation. After blocking of endogenous peroxidase for 20 min (6.3% v/v H$_2$O$_2$, 0.9% v/v methanol in PBS), washing in PBS, and antigen retrieval for 10 min at 92 °C (10 mM sodium citrate; pH 6.0), slides were briefly washed in TBS-T and unspecific binding sites were blocked for 2 h (10 % v/v NDS/NGS in TBS-T). Sections were incubated with primary antibodies (in 5% v/v NDS/NGS in TBS-T) overnight at 4 °C. Primary antibodies were detected with biotinylated secondary antibodies (in 5% v/v NDS/NGS in TBS-T) using the avidin-biotin-complex method with DAB as chromogen (Vectastain Elite ABC Kit Peroxidase, #VEC-PK-6100-KI01, Vector Laboratories). Color development was followed under a microscope and stopped by immersion of the sections in tap water. Sections were mounted in Aquatex® (#1.08562.0050, VWR). For control sections, the primary antibody was omitted. Normal sera used for blocking matched the species of the secondary antibody. Stained sections were documented using the microscope setup described under "X-gal staining".

Collagens were detected by picrosirius red staining. Sections of the aortic arch were deparaffinized and rehydrated and then incubated with sirius red staining solution (0.1% Direct Red 80 w/v in saturated picric acid solution) for 1 h in the dark, washed twice in 0.5% (v/v) acetic acid, dehydrated, and mounted in DPX (#06522, Sigma-Aldrich). Stained sections were documented using the microscope setup described under "X-gal staining" with a minor modification. Instead using normal bright field illumination, sections were imaged under cross-polarized light by adding two polarization filters in the light path. Under cross-polarized light, picrosirius red stained thick collagen fibers appear red-orange due to their birefringence.

Calcification of atherosclerotic plaque sections was detected by staining calcium with Alizarin Red S. Sections of the aortic arch were deparaffinized and rehydrated and then incubated with alizarin red staining solution (2% Alizarin Red S w/v in deionized water, pH 4.2) for 5 min. Excess dye was shaken off, sections were washed in acetone, dehydrated, and mounted in DPX. Stained sections were documented using the microscope setup described under "X-gal staining", and calcium deposits (orange-red) were counted by visual inspection.

The αSMA+ area of atherosclerotic plaques in immunohistochemical stainings was quantified with Fiji. First, individual plaques on sections of the aortic arch were marked with a polygon selection and saved as regions of interest (ROIs). As border between plaque and media, the first elastic lamina below the plaque that was not disrupted was chosen. Each plaque region was subdivided in cap and core region. The cap region was defined as the 30-μm thick layer at the luminal side of a plaque as suggested by others[70]. The core region was defined as the plaque without the cap region. Images were white balanced using the BIOP SimpleColorBalance plugin, and then Fiji's color deconvolution (option: "H DAB") was used to extract the stained area from the image (DAB channel). The DAB channel was thresholded to remove background from the staining and converted to a mask. Fiji's particle analyzer was used to mark the αSMA+ area in the image and the αSMA+ area was quantified in the whole plaque, cap, and core region. Finally, the αSMA+ area was normalized to the respective whole plaque, cap, and core area.

The collagen+ area was quantified as described for the αSMA+ area with minor adaptations. The plaque area was not subdivided in cap and core region. No white balance was performed. Instead of color deconvolution, images were split in their red, green, and blue channels in Fiji. The red channel (thick collagens) was thresholded instead of the "DAB channel".

To quantify the number of GC-B LacZ reporter positive cells, 10-μm paraffin sections of X-gal-stained aortae were analyzed before deparaffinization. The same microscope setup as described under "X-gal staining" was used for counting X-gal+ cells. All positive cells of the aortic arch (including branching vessels) were counted manually under the microscope and grouped in X-gal+ cells residing in the media and plaque, respectively. The total number of X-gal+ cells was normalized to the number of sections containing at least one X-gal+ cell.

## Western blot analysis

For Western blot analysis of protein expression, VSMCs were grown in 10-cm cell culture dishes (Corning) for 6–11 days until they reached ≥80% confluency. Cells were serum-starved overnight and lysed using a cell scraper and SDS-containing buffer (21 mM Tris-Cl, 0.67% w/v SDS, 0.2 mM PMSF). For analysis of protein expression in atherosclerotic aortae, aortae were carefully cleaned from surrounding tissue, and the plaque-rich regions (aortic arch and abdominal aorta) of two aortae were pooled and quick-frozen in liquid nitrogen. The tissue was lysed using tissue lysis buffer (50 mM Tris-Cl, 100 mM NaCl, 5 mM EDTA, 2% w/v SDS, 2.5 mM PMSF) and a FastPrep-24 device (MP Biomedicals). Lysates of cultured VSMCs and aortae were denatured at 95 °C for 10 min and centrifuged 5 min at 18,000 rcf at 4 °C. The protein concentration of the supernatant was determined using a total protein kit (Micro-lowry, Peterson's modification, #TP0300-1KT Sigma-Aldrich). Equal amounts of protein were loaded per gel and blotted onto a PVDF membrane (#03010040001, Roche) using a semi-dry blotting device (Trans-Blot® SD semi-dry transfer cell, Bio-Rad). Proteins immobilized on the membrane were detected by indirect chemiluminescence using HRP-coupled secondary antibodies and the WesternBright® Sirius kit (#K-12043-D20 Biozym). Chemiluminescence was detected with the ChemiDoc MP (Bio-Rad) documentation system. All blots shown in this paper are also provided as uncropped and unprocessed images in the source data file.

## cGMP/FRET imaging of cultured VSMCs and isolated aortae

For cGMP/FRET measurements of cultured VSMCs, cells were isolated from cGMP sensor mice expressing the cGMP biosensor cGi500. VSMCs were cultured on 12 mm or 25 mm glass coverslips for 5–13 days and then serum-starved for 12–24 h before the imaging experiment. To record ratiometric cGMP/FRET signals at the single-cell level, the Axiovert 200 microscope setup used for imaging of immunofluorescence staining (see, "Preparation and immunofluorescence staining of VSMC cultures") was modified by mounting a beam splitter with 05-EM insert (Micro-Imager DUAL-View, Photometrics) in front of the CCD camera to separate CFP and YFP emission for simultaneous recording. The beam splitter was composed of a 516 DCLP mirror and two emission band pass filters (BP 480/50 and BP 535/50). Furthermore, a superfusion system was used to superfuse the cells constantly with imaging buffer (140 mM NaCl, 5 mM KCl, 1.2 mM MgCl$_2$, 2 mM CaCl$_2$, 5 mM HEPES, 10 mM D-glucose, pH 7.4) during the measurement. The superfusion system consisted of an FPLC pump (Pharmacia P-500, GE Healthcare), FPLC injection valves (Pharmacia V-7, GE Healthcare), a vacuum pump (Laboport N86, KNF Neuberger), a 2 mL sample loop, tubing (Tygon S3 E-3603, Saint-Gobain), and a superfusion chamber for 12 mm coverslips (RC-25, #64-0232, Warner Instruments) attached to a magnetic chamber holder (PM-1, #64-1526, Warner Instruments). During measurements, cells were superfused with a flow rate of 1 mL/min at room temperature. Sensor fluorescence

was excited at 445 nm for 300 ms and recorded at 480 nm (CFP) and 535 nm (YFP) at 0.2 Hz with 4 × 4 binning. Drugs were applied via the 2 mL sample loop for a defined period without interrupting the measurement.

For cGMP/FRET measurements of aortae ex vivo, acutely isolated aortae were cut into 3–4 mm long pieces, opened longitudinally, and measured from the luminal side using a confocal spinning disk setup (Visitron). The microscope was controlled by the VisiView® software for image acquisition (Visitron). An upright microscope (Examiner.Z1, ZEISS) in combination with a spinning disk unit (CSU-X1, Yokogawa) was equipped with an EC Plan-NeoFluar 2.5x/0.085 air objective (ZEISS) and two water immersion objectives (W Plan-Apochromat 20x/1.0 and W Plan-Apochromat 40x/1.0, ZEISS). For excitation of fluorophores, two diode lasers were used (445 nm 100 mW and 488 nm 100 mW). To record high resolution images, a monochrome CCD camera (SPOT Pursuit Monochrome, SPOT Imaging) was used in combination with the following di-/polychroic mirrors and emission filters (Chroma) T405/488/568/647 + ET525/50 (YFP). For detection of cGMP/FRET signals, a sensitive EM-CCD camera (QuantEM 512SC, Photometrics) was used. Like in cell culture measurements, a beam splitter (Dual-View DV2) with insert (Micro-Imager DUAL-View, Photometrics) was used to allow simultaneous detection of CFP and YFP emission. It contained a dichroic mirror (505 DCLP) and two emission filters (BP 470/24 and BP 535/30). A superfusion system similar to the system described above for cGMP imaging of cultured VSMCs was used. However, instead of the superfusion chamber RC-25, the chamber RC-26 (#64-0234, Warner Instruments) was used. This chamber was sealed from below with a 24 × 40 mm square coverslip. It was wide enough for the water immersion objectives to enter from above. Healthy aortae were fixed on the coverslip using a block of Sylgard 184 with needles, and atherosclerotic aortae with a mesh and a slice hold-down (SHD-26H/10, Warner Instruments). During measurements, tissues were superfused with imaging buffer at 37 °C with a flow rate of 2 mL/min. Sensor fluorescence was excited at 445 nm for 200 ms and recorded at 470 nm (CFP) and 535 nm (YFP) at 0.2 Hz with a gain of 500 without binning. Drugs were applied via the 2 mL sample loop for a defined period without interrupting the measurement. After the recording, the three-dimensional structure of the vessel wall was documented at high resolution by acquisition of Z-stacks using the SPOT camera. To represent the 3D structure of plaques in videos, the "3D Project" function of Fiji was used. To represent the 3D structure of plaques in 2D, the "Z Project" function with "Max Intensity" option was used to generate a maximum intensity projection.

## Offline analysis of cGMP/FRET measurements and classification of ANP-/CNP-preferring cells

The images that were acquired during measurements were analyzed with Fiji to extract the mean fluorescence intensity of ROIs/cells. To reduce disturbing effects of tissue movement over time, the images from measurements of isolated aortae were aligned using the Multi-StackReg plugin in Fiji before fluorescence intensities were retrieved. cGMP/FRET signals were calculated using MS Excel (Microsoft Corporation) and analyzed with Origin Pro software (OriginLab Corporation). First, the CFP emission (F480, cyan in respective graphs) and YFP emission (F535, yellow in respective graphs) were corrected for background signals (mean fluorescence intensity of a cell-free region). Then, the cGMP/FRET signals (CFP/YFP ratio R ∼ [cGMP]) were calculated and normalized to the baseline (first 20–30 frames of a measurement). Changes of the ratio traces were considered as cGMP changes if the individual CFP and YFP channels showed antiparallel movement (see, ref. 98. for a detailed description of the working principle of cGi500). If no antiparallel movement could be confirmed (e.g., due to strong focus drifts in ex vivo measurements), signals were excluded from analysis. The quality of ratio traces was determined by visual inspection and low-quality ratio traces were discarded (e.g.,

highest signal smaller than twice the baseline noise, extreme "random" movements of the baseline).

To investigate whether a cultured VSMC responded more strongly to ANP or to CNP, the baseline drift of ratio traces was corrected using an exponential function, if possible. Otherwise, a linear baseline correction by interpolation was used. The peak height, reflecting the maximal cGMP response, was determined using the "Peak Analyzer" function of Origin Pro on smoothed ratio traces. Cells that responded at last 1.5 times stronger to ANP than to CNP at the same concentration (50 nM or 250 nM) were classified as ANP-preferring cells. Cells that responded at last 1.5 times stronger to CNP than to ANP at the same concentration (50 nM or 250 nM) were classified as CNP-preferring cells. Cells that responded similarly to ANP and CNP without showing a preference were classified as ANP ~ CNP cells. To compare the strength of NO-induced cGMP responses between ANP- and CNP-preferring VSMCs, the peak height of the NO-induced cGMP response was normalized to the highest peak elicited by ANP or CNP, respectively.

In ex vivo measurements of isolated aortae, we frequently observed changes in the ratio trace due to tissue movement, confounding quantitative evaluation of an overlapping cGMP response. Therefore, we performed only qualitative assessment whether or not a given stimulus triggered cGMP formation in a particular ROI/cell. Only unambiguous antiparallel movements of the individual YFP and CFP channels were scored as cGMP responses. To calculate the proportion of ROIs/cells that responded to ANP, CNP or DEA/NO, all ROIs/cells that responded to the respective stimulus with a cGMP/FRET change were counted and normalized to all ROIs/cells that responded to at least one of these three stimuli.

### Correlation of cGMP response patterns with protein expression in individual VSMCs

For correlation experiments, VSMCs were grown on gridded glass coverslips (Photoetched Cover Slips 12 mm round, Dunnlab, #1916-91012). These coverslips are provided with a coordinate grid that can be seen under bright field illumination and thus enabled the identification of the same region of cGMP/FRET measurements and subsequent immunofluorescence staining. First, cGMP/FRET measurements were performed, and the coordinates of the measured region were documented with a bright field image. The cells were then fixed and immunostained. The region previously analyzed in the cGMP measurement was identified by its known coordinates on the immunostained coverslip and another brightfield image was taken. Further steps were performed using Fiji and MS Excel. The images of the coordinates were used to align the immunostained cells with the cells recoded during the cGMP/FRET measurement. This "mapping method" allowed us to correlate the cGMP response and protein expression pattern in individual VSMCs. A detailed description of the alignment procedure is shown in Supplementary Fig. 2. The mean fluorescence intensity of each cell in the immunofluorescence staining for a particular protein was corrected for background fluorescence (fluorescence signal in a cell-free region) and correlated with the corresponding cGMP response pattern of the same cell.

### Analysis of VSMCs by scRNA-seq

Primary aortic VSMCs were isolated from wildtype mice as described under "Preparation and immunofluorescence staining of VSMC cultures". 200,000 cells were plated per well of a 6-well plate (3 wells in total) and cultured in full medium at 37 °C and 6% $CO_2$. After 5 days in culture, cells were detached with trypsin/EDTA. After addition of full medium and centrifugation, cells were resuspended in PBS with 0.04% BSA and passed through a 30-µm filter. The number of viable cells was determined using an automated cell counter. 10,000 live cells were loaded onto a 10x Genomics microfluidic chip and encapsulated with barcoded oligo-dT-containing gel beads using 10x Genomics Chromium controller according to the manufacturer's instructions. Single-cell libraries were then constructed according to the manufacturer's instructions. Libraries were sequenced on a NextSeq 500/550 system (Illumina).

To analyze the scRNA-seq data, Fastq files were aligned to the reference genome using CellRanger Software (10x Genomics). The processed sequencing data was analyzed using the R package Seurat (latest version 5.1.0). Cells with <300 genes, <3000 RNAs, or ≥20% mitochondrial RNAs were considered of poor quality and removed from the dataset. Genes that were detected in fewer than ten cells were also removed from the dataset. After quality control, the reads were normalized and scaled using the SCTransform function of the Seurat R package with standard parameters. Principal component analysis was performed on the 3000 most variable genes and the first 40 principal components were used for clustering and for two-dimensional visualization of the resulting clusters by Uniform Manifold Approximation and Projection (UMAP). For cell type identification and subsequent analyses, the gene expression values were normalized using the "NormalizeData" function with standard parameters.

To analyze *Npr1* and *Npr2* expression in published murine and human scRNA-seq data, we downloaded the following processed datasets from the PlaqView portal (https://www.plaqview.com/): "Alencar_2020_dual" (mouse, ref. [79]), "Pan_2020_mouse" (mouse, ref. [78]), "Pan_2020" (human, ref. [78]). The processed sequencing data was analyzed using the R package Seurat. The authors' cell type annotations have been adopted. Cell clusters that were not relevant for our study (e.g., immune cells) were removed from the datasets, and UMAPs/Feature plots were created. The percentage of *Npr1/2* expressing cells per cluster was extracted from the "data" slot provided by the DotPlot function.

### Statistical analysis and reproducibility

Statistical analysis was performed in Origin Pro. All tests were performed two-tailed. The Shapiro-Wilk test was used to test data for normal distribution. The Mann–Whitney U test was performed for pairwise comparisons of non-normally distributed data. Normally distributed data was tested for homo-/heteroscedasticity (F-test). A standard Student's t-test (homoscedastic data) or a Welch's t-test (heteroscedastic data) was performed for pairwise comparisons. For groupwise comparisons of more than two groups, one-way analysis of variance (ANOVA) with subsequent Bonferroni post hoc test or a Kruskal-Wallis ANOVA was performed. Alternatively, multiple pairwise comparisons (Student's t-test or Mann–Whitney U test) were performed in combination with a Bonferroni correction for multiple comparisons. Pearson's chi-squared test was used to analyze whether categorical data was distributed differently between conditions. Statistical significance was considered for $p < 0.05$ and indicated in three categories (*$p < 0.05$, **$p < 0.01$, ***$p < 0.001$). The detailed statistical analysis is provided in the Source Data file. Sample sizes were chosen based on data from previous publications, which were sufficient for statistical analysis. For cGMP/FRET and immunofluorescence analysis, few cells were excluded based on pre-defined criteria, which are stated in the respective figure legends. The number of replications is stated in the figure legends. The experiments were not randomized. Experimenters were blinded to sex and genotype.

### Reporting summary

Further information on research design is available in the Nature Portfolio Reporting Summary linked to this article.

## Data availability

The raw scRNA-seq data of cultured primary aortic VSMCs generated in this study has been deposited in the Gene Expression Omnibus (GEO) database under accession code GSE282944. The processed version of this data is available in the PlaqView portal (https://www.plaqview.com/) under the accession code Lehners_2024. The scRNA-

seq data shown in Supplementary Fig. 9 is available in the PlaqView portal (https://www.plaqview.com/) under accession codes Alencar_2020_dual, Pan_2020_mouse, and Pan_2020 (see Source Data file for a detailed description). Source data are provided with this paper.

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

## Acknowledgements

We thank Barbara Birk for technical assistance, Malte Roessing for support of the atherosclerosis experiments, Michael Paolillo for help with cGMP imaging, Hyazinth Dobrowinski for contributing to some cell culture experiments, Andreas Friebe for the NO-GCβ1 antibodies, and our animal facility for the care of mice. Special thanks go to all members of the Feil laboratory including lab students for experimental support and constructive discussions. This work was supported by grants from the EU Framework Program for Research and Innovation "Horizon 2020"–ERA-CVD JTC2017-044 (S.F.), the Deutsche Forschungsgemeinschaft (German Research Foundation) Projektnummer 335549539-GRK 2381 (R.F., S.F., H.S., and R.L.), Projektnummer 464254052-GRK 2816 (R.F.), and FOR 2060 projects FE 438/5-2 (R.F.), FE 438/6-2 (R.F.) and LU 1490/3-2 (R.L.), and the Dr. K.H. Eberle Stiftung (R.F. and S.F.), Reinhard Frank-Stiftung (R.F.) and Elisabeth und Franz Knoop-Stiftung (D.S.).

## Author contributions

M.L. performed most of the experiments and was strongly involved in experimental design, data analysis, and manuscript writing. H.S. and S.F. provided essential reagents and mouse models. M.Z. performed and analyzed scRNA-seq experiments. D.S. and M.K. supported the cell culture experiments. A.P. analyzed serum lipid profiles. J.A. and R.L. performed carotid ligation. All authors provided constructive suggestions and edited the manuscript. R.F. directed the study and wrote most of the manuscript.

## Funding

## Competing interests

The authors declare no competing interests.
