## [Peer review File · Nature Communications]

REVIEWER COMMENTS

Reviewer #1 (Remarks to the Author):

Phenotypic regulation of vascular smooth muscle cells (VSMCs) is central to vascular disease and there is a growing interest in identifying ways to target these cells clinically. cGMP signaling is an attractive candidate for this purpose, given the successful development of specific drugs to manipulate this pathway. cGMP mediates VSMC relaxation and has been associated with regulation of VSMC plasticity in vascular disease (e.g. doi's 10.1038/s42003-022-03140-2, 10.3390/jcdd5020020, 10.1038/s41467-021-22933-3). This study aims to improve our understanding of how the pathway is regulated in the light of VSMC heterogeneity, which will be valuable to improve efficacy and specificity.

The Feil group has pioneered technical developments to study the activity of the pathway, using a genetic approach to generate an *in vivo* cGMP biosensor, which has been used in other tissues and is here applied in VSMCs.

The study provide data demonstrating heterogeneity among cultured VSMCs in the response to guanylyl cyclases (GC) peptide ligands, and show that increased sensitivity to CNP vs ANP correlates with a more de-differentiated VSMC phenotype. The data from cultured cells is convincing, although experimental details are not clear (e.g. regarding how the cells analyzed were distributed between biological replicates and how many cells were excluded from analysis). Evidence that differential sensitivity is due to changes in expression of GC isoforms with different ligand preference is provided using single cell analysis of cultured VSMCs. It would have been useful to understand how these factors are regulated in disease, which could be done using the variety of scRNA-seq datasets from both experimental atherosclerosis and human plaques.

Imaging of arteries from mice expressing the cGMP sensor provides important physiological evidence that a switch from ANP/GC-A to CNP/GC-B also occurs in disease. The analysis of mice only expressing the sensor in VSMCs is important provided the variety of cell types within atherosclerotic lesions. However, the impact of cell scoring by imaging depth is not clear, which is important given the differences between cells in different plaque regions (fibrous cap, shoulder, necrotic core). It would also be helpful for interpretation to understand the activity of the sensor in the different VSMC types described (as per comment on scRNA-seq above).

The implications of the switch from ANP/GC-A to CNP/GC-B usage is explored by genetic ablation of GC-B. This part of the manuscript is less developed and support for the conclusion of "Markedly altered plaque composition when GC-B was ablated in VSMCs" is not entirely convincing. For

example one of the read-outs for VSMC changes is the activity from the beta-galactosidase inserted in the GC-B locus – yet the in vitro data suggests that X-gal staining only detects a subset of GC-B-positive cells, but this is not better characterized. Furthermore, the scoring of lesion parameters is not transparent, e.g. the relevance of comparisons within genotype was not clear to me, and a small number of animals are analysed. However, my main concern is that the analysis does not provide solid information about how the biology of VSMCs is perturbed. For examples, would ablation of CNP sensitivity affect the proliferation of cells, and/or the ability to change into one of the many sub-types of plaque VSMCs that have been described. It is also not clear whether other plaque cell types are changed, for example the *Lgals3* is expressed by both de-differentiated VSMCs and macrophages. This information would help understanding how targeting of the pathway would impact on disease.

Therefore, while the observation that VSMC GC ligand preference is altered in vascular disease is interesting and relevant for targeting this pathway in patients, further evidence for how manipulation of this axis would impact on VSMC regulation is needed for this to be of interest for a general readership.

Reviewer #2 (Remarks to the Author):

The study by Lehnert et al performed an impressive single-cell analysis of mouse vascular smooth muscle cells (VSMCs) under healthy and diseased conditions (atherosclerosis model) to dynamically assess the activity of the three cGMP generating pathways – via NO and the two natriuretic peptide receptors GC-A (receptor for ANP/BNP) and GC-B (receptor for CNP). This is an impressive study which could indeed assess the heterogeneity of VSMC phenotypes in vitro and in vivo by a groundbreaking combination of techniques- live cell imaging of cGMP, direct correlation with phenotypic VSMC marker expression by immunofluorescence in the same cells, and scRNAseq. Strikingly, the three previously described phenotypes of contractile, synthetic and modulated VSMCs identified by single-cell sequencing were found to be related to the activity of ANP/GC-A and CNP/GC-B signaling, with a switch from ANP- and NO-dependent cGMP generation in contractile VSMCs to CNP/GC-B-cGMP signaling during phenotypic modulation both in cell culture and most importantly in the context of atherosclerotic plaque remodeling (showing an increased number of the chondrocyte-like cells with high GC-B expression and CNP responsivity). GC-B receptor expression, which was simultaneously tracked in these cells using lacZ knockin into its genetic locus, was found to be a marker for VSMC modulation and atherosclerosis development, with the cap-to-core ratio of α SMA positive cells significantly changed due to phenotypic modulation in GC-B knockout mice. In general this provides a model of atheroprotective CNP/GC-B

signaling in VSMCs which attenuates the development of potentially harmful VSMC-derived plaque cells. In general, this is a highly novel and impressive piece of work shown in a extremely well written manuscript. Before publication I would recommend the authors to address the following points.

1. Lots of VSMC data are reported for indicated numbers of measured/analyzed VSMCs given typically as a range, e.g. n = 20-42 cells. I would recommend to include the number of independent cell isolations for in vitro experiments and a number of mice for in vivo/ex vivo imaging studies which should be indicated in the figure legends.

2. Several of the FRET study figures (fig 1c, suppl fig S4, suppl fig S7b/c) show representative traces but lack quantification of response amplitudes or total/relative number of cells responding or not responding to given stimuli. This could be added to the figures to improve data presentation.

3. All experiments performed in this study are done on mice. How relevant are these finding to a human situation? Please discuss. Indeed, in the discussion the authors mentioned that the GC-A/GC-B switch was also found in human atherosclerotic lesions (ref 76) in terms of expression. If there any possibility to perform scRNAseq or analyse available datasets from human samples for the expression GC-A/GC-B and other cGMP pathway components.

Reviewer #3 (Remarks to the Author):

Single-cell analysis identifies the CNP/GC-B/cGMP axis as a marker and regulator of modulated VSMCs in atherosclerosis

In this manuscript, Lehnert et al. conduct a detailed analysis of murine Vascular Smooth Muscle Cells (VSMCs) under stimulation, specifically examining natriuretic peptide signaling pathways at the single-cell level. The study reveals that the ANP/GC-A and CNP/GC-B pathways are primarily active in contractile and modulated VSMCs, respectively. Additionally, the authors identify the CNP/GC-B/cGMP axis as a new marker and regulator of dedifferentiated chondrocyte-like VSMCs in the context of atherosclerosis. However there are a few concerns, addressing which would further strengthen this work:

“...the morphology of most ANP-preferring VSMCs was elongated, whereas CNP-preferring cells were more roundish (Fig. 1c, right picture, orange and cyan cell borders, respectively).” Could the authors please provide a more quantitative analysis of morphological differences between the two sub-populations?

“GC-B knockout VSMCs responded to ANP and/or NO, but not to CNP, demonstrating that CNP-induced cGMP generation was exclusively mediated by GC-B (Fig. 1d).” A more accurate conclusion from this observation would be that “GC-B exclusively participates in CNP induced cGMP generation and not ANP/NO.” As there could be other mediators of CNP induced cGMP generation not tested by this specific experiment. Could the authors please clarify?

Could the authors comment on the stability of ANP vs CNP preference, is this bias maintained over time? Do daughter cells of an ANP preferring cell also have the same bias? Is this something the authors can measure using their existing technical approach?

In supplementary fig 2b, I was surprised to see that the differences in α SMA/SM22a between ANP vs CNP preferring cells wasn't as stark. For instance, many CNP and ANP preferring cells have roughly the same α SMA expression level. Could the authors please comment on that? Is this an artifact of the fluo measurement/ potential incorrectly classified cells or something else?

Fig. 2 - it was unclear to me if “ Data are shown as mean + SEM (n = 97-112 cells)” mentioned in the legend was only for 2f or all barplot panels.

In contrast to the discrete classification of cells as ANP vs CNP preferring, is there a continuous gradient of cells in terms of preference, ranging from ANP to CNP. Since all cells not showing a strong preference either way are grouped into the ANP~CNP group, it is hard to assess whether that is the case or not. And if so, do other characteristics (e.g. morphological differences) also lie on a similar gradient? This could perhaps be assessed with the single cell measurements that the authors have already collected for these experiments.

“Comparison of the top cell type-specific transcripts showed that the pSMC cluster was more similar to the mSMC cluster than to the cSMC cluster (Fig. 3b).” This needs to be backed by more quantitative inspection of the extent of similarity (e.g. cluster/ cell type level correlations).

“Indeed, $\approx 24\%$ of the mSMCs co-expressed *Npr2* and *Tnfrsf11b* mRNA (Fig. 3e yellow dots).” Could the authors please provide more details on how co-expression was defined in the single cell analysis?

In Fig. 4c,d the total cell counts are quite low, could the authors perform a statistical test (e.g. randomization test) to assess if the compositional changes reported are statistically significant?

“...only the outer cell layers of an atherosclerotic plaque, but not the inner core region, could be measured with sufficient intensity to evaluate cGMP signals...” Could the authors please comment on the implications of this limitation on their results?

Reviewer #4 (Remarks to the Author):

Response to Reviewer 1

Phenotypic regulation of vascular smooth muscle cells (VSMCs) is central to vascular disease and there is a growing interest in identifying ways to target these cells clinically. cGMP signaling is an attractive candidate for this purpose, given the successful development of specific drugs to manipulate this pathway. cGMP mediates VSMC relaxation and has been associated with regulation of VSMC plasticity in vascular disease (e.g. doi's 10.1038/s42003-022-03140-2, 10.3390/jcdd5020020, 10.1038/s41467-021-22933-3). This study aims to improve our understanding of how the pathway is regulated in the light of VSMC heterogeneity, which will be valuable to improve efficacy and specificity.

The Feil group has pioneered technical developments to study the activity of the pathway, using a genetic approach to generate an *in vivo* cGMP biosensor, which has been used in other tissues and is here applied in VSMCs.

1) The study provide data demonstrating heterogeneity among cultured VSMCs in the response to guanylyl cyclases (GC) peptide ligands, and show that increased sensitivity to CNP vs ANP correlates with a more de-differentiated VSMC phenotype. The data from cultured cells is convincing, although experimental details are not clear (e.g. regarding how the cells analyzed were distributed between biological replicates and how many cells were excluded from analysis).

We have updated the figure legends with more information about the experimental details. For *ex vivo* measurements of blood vessels, we have added the number of mice/measured blood

vessels as well as the number of measured regions and recordings excluded from analysis. For *in vitro* recordings of VSMCs, the number of cell isolations, measured cover slips, recorded cells and cells excluded from analysis are now indicated for every panel. Note that the classification of cultured VSMCs with preference for ANP or CNP has been carried out in multiple independent experiments (e.g., shown in Figs. 1, 2, and 4; Suppl. Figs. 1, 3, and 4). In total, we have analyzed >2400 cells in 45 measurements from global cGi500 sensor mice and >370 cells in 17 measurements from SMC-specific cGi500 sensor mice (see Suppl. Fig. 3c).

2) Evidence that differential sensitivity is due to changes in expression of GC isoforms with different ligand preference is provided using single cell analysis of cultured VSMCs. It would have been useful to understand how these factors are regulated in disease, which could be done using the variety of scRNA-seq datasets from both experimental atherosclerosis and human plaques.

This is certainly an important point. In the original manuscript, we have manually analyzed the scRNAseq data published in three relevant studies (refs. 77, 78, 79; see page 17, last para): “Our analyses of atherosclerotic lesions are also in agreement with recent RNA sequencing studies. Mokry et al. reported GC-B expression in so-called fibro-inflammatory human plaques⁷⁷, and Pan et al. and Alencar et al. detected GC-B mRNA in mouse and human plaques specifically in populations of modulated VSMCs classified as fibrochondrocytes⁷⁸ or chondrocyte-like cells⁷⁹.”

We now used the PlaqView portal (<https://www.plaqview.com/>) to analyze murine and human scRNAseq datasets published in the above referenced papers and other studies in more detail. If we understand correctly, the reviewer suggested comparing GC isoform expression in healthy vs diseased (atherosclerotic) arteries. Unfortunately, we could not find such matched scRNAseq datasets with appropriate annotation in the repository. However, using PlaqueView we were able to access Npr1 (GC-A) and Npr2 (GC-B) mRNA expression in various cell populations of murine and human atherosclerotic lesions (see figure for the reviewer). Note that we were not able to access the human data published by Mokry et al. (ref. 77). In agreement with our own findings, Npr2 mRNA was enriched in the murine fibrochondrocyte cluster, while Npr1 was strongly expressed in contractile SMCs and downregulated in fibrochondrocytes. In the human dataset, Npr2 mRNA expression appeared to be relatively low but detectable, and the distribution of Npr2 vs Npr1 was similar to murine lesions. In addition to the statement in the main text (page 17, last para), we have now included a new table summarizing the percentages of GC-A (Npr1) and GC-B (Npr2) positive contractile and modulated SMCs detected in the above-mentioned murine and human datasets (Suppl. Table 2).

FIGURE FOR THE REVIEWER: Characterization of Npr1 and Npr2 expression in published murine and human scRNA-seq datasets using PlaqView. UMAP visualization of various cell clusters from atherosclerotic plaques (left in each panel) and their Npr1 and Npr2 expression (right in each panel). Two murine (a, b) and one human (c) datasets are shown. The expression level is illustrated by a color scale. Each dot represents a single cell. Red circles highlight transdifferentiated SMC clusters with increased Npr2 and decreased Npr1 expression.

3) Imaging of arteries from mice expressing the cGMP sensor provides important physiological evidence that a switch from ANP/GC-A to CNP/GC-B also occurs in disease. The analysis of mice only expressing the sensor in VSMCs is important provided the variety of cell types within atherosclerotic lesions. However, the impact of cell scoring by imaging depth is not clear, which is important given the differences between cells in different plaque regions (fibrous cap, shoulder, necrotic core).

The reviewer is absolutely right. Due to limitations in the penetration depth of light, we could only measure the outer layers of atherosclerotic plaques. These measured regions were most likely the fibrous cap, shoulder, and upper layers of the plaque core. However, for technical reasons, it was difficult to clearly assign a recorded FRET/cGMP signal to one of these specific plaque regions. To clarify this, we modified the respective text from *“However, only the outer cell layers of an atherosclerotic plaque, but not the inner core region, could be measured with sufficient intensity to evaluate cGMP signals”* (page 12, 1st para) to *“However, only the outer*

cell layers of an atherosclerotic plaque (presumably the fibrous cap, shoulder, and upper region of the plaque core), but not the inner core region, could be measured with sufficient intensity to evaluate cGMP signals”.

The technical limitations of optical imaging described above meant that we were unable to characterize cells in the inner core of the plaque by cGMP/FRET imaging. However, please note that we also used the genetic GC-B lacZ reporter as an alternative method to detect GC-B-expressing plaque cells. This approach did not have the aforementioned limitations of *ex vivo* cGMP/FRET imaging of atherosclerotic lesions and clearly showed that cells with high GC-B expression exist in the plaque core (Fig. 6g, i). We added a respective statement to the discussion section (page 17, end of 1st para).

4) It would also be helpful for interpretation to understand the activity of the sensor in the different VSMC types described (as per comment on scRNA-seq above).

The cGi500 biosensor construct used in this study was integrated into the Rosa26 locus. Its expression was driven by the ubiquitous CAG promoter. This configuration usually leads to widespread/global expression of the transgene in most cell types and tissues. Indeed, we detected sensor expression basically in all tissues and cells of our primary VSMC cultures. As we could detect cGMP signals in ANP-preferring cells, ANP~CNP cells, and CNP-preferring cells in culture, we are confident that the biosensor was expressed and active in the different types of VSMCs analyzed in our study. Furthermore, sensor expression was detected throughout the atherosclerotic plaque regions analyzed by us. The inability to detect sensor-expressing cells in the core of atherosclerotic lesions was most likely due to technical limitations like insufficient penetration of light and/or light scattering by accumulated lipids or cell debris. Considering that we could measure clear cGMP signals in strongly modulated cultured VSMCs, it is unlikely that problems with sensor expression *per se* excluded the recoding of cGMP signals from cells in the plaque core.

5) The implications of the switch from ANP/GC-A to CNP/GC-B usage is explored by genetic ablation of GC-B. This part of the manuscript is less developed and support for the conclusion of "Markedly altered plaque composition when GC-B was ablated in VSMCs" is not entirely convincing.

We phrased the sentence more carefully and changed it to *“However, immunostaining of aortic sections for marker proteins and X-gal staining for GC-B gene activity as a means of detecting modulated VSMCs showed an altered plaque composition when GC-B was ablated in VSMCs.”* (page 14, 1st para)

Note that we have now added more data on plaque composition demonstrating a role of GC-B in the control of an osteochondrogenic/fibrotic plaque state (see below, point 8, and new Figure 7).

6) For example one of the read-outs for VSMC changes is the activity from the beta-galactosidase inserted in the GC-B locus – yet the in vitro data suggests that X-gal staining only detects a subset of GC-B-positive cells, but this is not better characterized.

Yes, we noted that lacZ expression from the GC-B lacZ reporter allele appears to be relatively low and does not detect all cells that have functional CNP-GC-B signaling. We have mentioned and discussed this limitation on page 11 (2nd para) and page 14 (2nd para). However, we used

the genetic lacZ reporter only as a complementary approach to live-cell cGMP imaging for monitoring GC-B positive cells. Indeed, the lacZ reporter turned out to be superior to cGMP imaging for detecting GC-B cells in the inner core of plaques, where cGMP imaging was not possible due to technical issues (see above, point 3). However, it would have gone beyond the scope of the present study to analyze the reasons for the relatively low sensitivity of GC-B detection via the genetic lacZ reporter.

7) Furthermore, the scoring of lesion parameters is not transparent, e.g. the relevance of comparisons within genotype was not clear to me, and a small number of animals are analysed.

Depending on the specific experiment shown in Fig. 6, between 5 and 21 mice/atherosclerotic aortas were examined and up to several hundred sections were evaluated. These numbers of animals or samples are commonly used for such experiments. A higher number of animals would not be readily approved by the authorities anyway.

The comparison of parameters within a genotype should show, for example, that, as expected, there were more SMA+ cells in the cap than in the core (Fig. 6e) or that the number of X-gal+ cells in the plaque of GC-B^{smko} but not control animals increased significantly compared to the media (Fig. 6h).

8) However, my main concern is that the analysis does not provide solid information about how the biology of VSMCs is perturbed. For examples, would ablation of CNP sensitivity affect the proliferation of cells, and/or the ability to change into one of the many sub-types of plaque VSMCs that have been described. It is also not clear whether other plaque cell types are changed, for example the Lgals3 is expressed by both de-differentiated VSMCs and macrophages. This information would help understanding how targeting of the pathway would impact on disease.

We have done additional work to address this concern and compared more features of atherosclerotic plaques from control and GC-B^{smko} mice. These data are presented in the new Fig. 7 and described in the manuscript on page 15, 2nd para.

The number of cell nuclei/plaque section did not significantly differ between GC-B^{smko} mice and control mice (new Fig. 7a). This finding is consistent with the similar lesion area in control and knockout mice (Fig. 6c) and suggests that ablation of GC-B in VSMCs does not have a strong effect on plaque cell proliferation and/or migration. However, our new data provide strong evidence that ablation of GC-B in VSMCs promotes a more fibrotic/osteochondrogenic plaque phenotype. On average, 42 % ± 5 % of X-gal+ (GC-B LacZ) cells in plaque sections were positive for the chondrocyte marker osteopontin (new Fig. 7b). Importantly, significantly more osteopontin+ cells as well as increased collagen deposition were detected in X-gal+ plaques of GC-B^{smko} mice as compared to control mice (new Fig. 7c-e). Moreover, while calcification of X-gal+ atherosclerotic lesions was rarely detected in control mice, we frequently found calcified plaques in X-gal+ aortic arch sections from GC-B^{smko} mice (new Fig. 7f, g).

Together, our new data indicate that ablation of GC-B in VSMCs promotes their trans-differentiation to bone-like plaque cells, also called “chondrocyte-like cells” or “fibro-chondrocytes”, without changing the absolute number of lesional cells. We believe that this information helps understanding how targeting of the pathway would impact on disease.

9) Therefore, while the observation that VSMC GC ligand preference is altered in vascular disease is interesting and relevant for targeting this pathway in patients, further evidence for how manipulation of this axis would impact on VSMC regulation is needed for this to be of interest for a general readership.

We believe that our new data (see above, point 8 and new Fig. 7) provides further evidence for how manipulation of the CNP-GC-B-cGMP axis would impact on VSMC regulation in atherosclerosis.

Response to Reviewer 2

The study by Lehnert et al performed an impressive single-cell analysis of mouse vascular smooth muscle cells (VSMCs) under healthy and diseased conditions (atherosclerosis model) to dynamically assess the activity of the three cGMP generating pathways – via NO and the two natriuretic peptide receptors GC-A (receptor for ANP/BNP) and GC-B (receptor for CNP). This is an impressive study which could indeed assess the heterogeneity of VSMC phenotypes *in vitro* and *in vivo* by a groundbreaking combination of techniques- live cell imaging of cGMP, direct correlation with phenotypic VSMC marker expression by immunofluorescence in the same cells, and scRNAseq. Strikingly, the three previously described phenotypes of contractile, synthetic and modulated VSMCs identified by single-cell sequencing were found to be related to the activity of ANP/GC-A and CNP/GC-B signaling, with a switch from ANP- and NO-dependent cGMP generation in contractile VSMCs to CNP/GC-B-cGMP signaling during phenotypic modulation both in cell culture and most importantly in the context of atherosclerotic plaque remodeling (showing an increased number of the chondrocyte-like cells with high GC-B expression and CNP responsivity). GC-B receptor expression, which was simultaneously tracked in these cells using lacZ knockin into its genetic locus, was found to be a marker for VSMC modulation and atherosclerosis development, with the cap-to-core ratio of α SMA positive cells significantly changed due to phenotypic modulation in GC-B knockout mice. In general this provides a model of atheroprotective CNP/GC-B signaling in VSMCs which attenuates the development of potentially harmful VSMC-derived plaque cells. In general, this is a highly novel and impressive piece of work shown in an extremely well written manuscript. Before publication I would recommend the authors to address the following points.

1) Lots of VSMC data are reported for indicated numbers of measured/analyzed VSMCs given typically as a range, e.g. n = 20-42 cells. I would recommend to include the number of independent cell isolations for *in vitro* experiments and a number of mice for *in vivo/ex vivo* imaging studies which should be indicated in the figure legends.

We have updated the figure legends with more information about the experimental details. For *ex vivo* measurements of blood vessels, we have added the number of mice/measured blood vessels as well as the number of measured regions and recordings excluded from analysis. For *in vitro* recordings of VSMCs, the number of cell isolations, measured cover slips, recorded cells and cells excluded from analysis are now indicated for every panel. Note that the classification of cultured VSMCs with preference for ANP or CNP has been carried out in multiple independent experiments (e.g., shown in Figs. 1, 2, and 4; Suppl. Figs. 1, 3, and 4). In total, we have analyzed >2400 cells in 45 measurements from global cGi500 sensor mice and >370 cells in 17 measurements from SMC-specific cGi500 sensor mice (see Suppl. Fig. 3c).

2) Several of the FRET study figures (fig 1c, suppl fig S4, suppl fig S7b/c) show representative traces but lack quantification of response amplitudes or total/relative number of cells responding or not responding to given stimuli. This could be added to the figures to improve data presentation.

We added the number of cells belonging to each category to the figure legend of Fig. 1c. Furthermore, the percentage of ANP-, ANP~CNP, and CNP-preferring cells summarized across all measurements are shown in Suppl. Fig. 3c. We now added to the figure legends of Suppl. Fig. 5 and Suppl. Fig. 8b/c, how many cells/regions of the respective measurement are represented by the traces.

We did not quantify the strength of responses in *ex vivo* measurements for two reasons. Firstly, *ex vivo* measurements were often perturbed by focus drifts potentially caused by tissue movement. These focus drifts could result in strong changes of the ratio traces and therefore did not allow for reliable quantification of response amplitudes. Nonetheless, we could still distinguish whether a change of the cGMP/ratio trace was caused by cGMP or tissue movements by verifying the antiparallel development of the CFP and YFP fluorescence traces. Secondly, the depth at which we measured in the tissue changed from measurement to measurement. So the concentration of each drug at the measured region would be different depending on the number of cell layers the drug had to pass through. Therefore, we assumed that the effective drug concentrations in different *ex vivo* measurements were not perfectly comparable.

3) All experiments performed in this study are done on mice. How relevant are these findings to a human situation? Please discuss. Indeed, in the discussion the authors mentioned that the GC-A/GC-B switch was also found in human atherosclerotic lesions (ref 76) in terms of expression. If there any possibility to perform scRNAseq or analyse available datasets from human samples for the expression GC-A/GC-B and other cGMP pathway components.

This is certainly an important point. It was also raised by reviewer 1. We now used the PlaqView portal (<https://www.plaqview.com/>) to analyze available murine and human scRNAseq datasets in more detail. In agreement with our own findings, Npr2 (GC-B) mRNA was enriched in the murine fibrochondrocyte cluster, while Npr1 (GC-A) was strongly expressed in contractile SMCs and downregulated in fibrochondrocytes. In the human datasets, Npr2 mRNA expression appeared to be relatively low but detectable, and the distribution of Npr2 vs Npr1 was similar to murine lesions (see new Suppl. Table 2). For more details including a figure for the reviewer, please see our response to reviewer 1, point 2.

We have now added a statement in the discussion regarding the need for more work on the relevance of our mouse studies for the human situation (page 18, end of 1st para): "Future studies should investigate whether CNP controls vascular and bone remodeling via similar or different molecular mechanisms and whether our findings in mice are also relevant for atherosclerosis in humans."

Response to Reviewer 3

Single-cell analysis identifies the CNP/GC-B/cGMP axis as a marker and regulator of modulated VSMCs in atherosclerosis

In this manuscript, Lehnert et al. conduct a detailed analysis of murine Vascular Smooth

Muscle Cells (VSMCs) under stimulation, specifically examining natriuretic peptide signaling pathways at the single-cell level. The study reveals that the ANP/GC-A and CNP/GC-B pathways are primarily active in contractile and modulated VSMCs, respectively. Additionally, the authors identify the CNP/GC-B/cGMP axis as a new marker and regulator of dedifferentiated chondrocyte-like VSMCs in the context of atherosclerosis. However there are a few concerns, addressing which would further strengthen this work:

1) “...the morphology of most ANP-preferring VSMCs was elongated, whereas CNP-preferring cells were more roundish (Fig. 1c, right picture, orange and cyan cell borders, respectively).” Could the authors please provide a more quantitative analysis of morphological differences between the two sub-populations?

We quantified the morphology of VSMCs from all three categories (ANP-preferring, ANP~CNP, CNP-preferring) for all cells of the measurement shown in Figure 1c using the roundness parameter of FIJI. This parameter ranges from “0” (elongated) to “1” (round). As shown in the new Suppl. Fig. 1b and discussed in the main text (page 8, 1st para), the roundness of cells increased significantly from ANP- over ANP~CNP to CNP-preferring cells.

2) “GC-B knockout VSMCs responded to ANP and/or NO, but not to CNP, demonstrating that CNP-induced cGMP generation was exclusively mediated by GC-B (Fig. 1d).” A more accurate conclusion from this observation would be that “GC-B exclusively participates in CNP induced cGMP generation and not ANP/NO.” As there could be other mediators of CNP induced cGMP generation not tested by this specific experiment. Could the authors please clarify?

Thank you for this comment. We tried to clarify this and revised the statement to “To confirm that CNP-induced cGMP signals were indeed generated via GC-B, we analyzed GC-B knockout VSMCs (Fig. 1d). These cells responded normally to ANP and NO, showing that the respective cGMP-generating systems were not impaired in the absence of GC-B. Their responsiveness to ANP indicated that the cells expressed GC-A, which in principle could also mediate CNP-induced cGMP increases. However, GC-B knockout VSMCs did not respond to CNP at all, demonstrating that CNP-induced cGMP generation was exclusively mediated by GC-B (Fig. 1d).” (page 7, end of 2nd para)

3) Could the authors comment on the stability of ANP vs CNP preference, is this bias maintained over time? Do daughter cells of an ANP preferring cell also have the same bias? Is this something the authors can measure using their existing technical approach?

The reviewer raised an interesting question. To answer this question, it would be necessary to establish longitudinal cGMP/FRET-measurements of individual VSMCs and their daughter cells over several days. Establishing these measurements would certainly be a challenging task and take some months. We plan to address this question in the future.

4) In supplementary fig 2b, I was surprised to see that the differences in aSMA/SM22a between ANP vs CNP preferring cells wasn't as stark. For instance, many CNP and ANP preferring cells have roughly the same aSMA expression level. Could the authors please comment on that? Is this an artifact of the fluo measurement/ potential incorrectly classified cells or something else?

Usually, our VSMCs were grown for 5-6 days before analysis. However, in this particular measurement, cells were grown for a relatively long time period (9 days) before analysis. As phenotypic modulation and the shift from ANP-preferring (aSMA high) to CNP-preferring (aSMA low) cells is a continuous process that progresses over time (see also point 6), it is possible that some of the ANP-preferring cells had already downregulated aSMA, while still responding preferentially to ANP. Indeed, in a measurement after 6 days in culture with the same SMC-specific cGMP reporter line, the differences in aSMA expression were stronger (see figure for the reviewer below). Since the expression of SM22a was not examined in this experiment, we have not presented it in the manuscript.

FIGURE FOR THE REVIEWER: Expression of α SMA in primary VSMCs from SMC-specific cGMP sensor mice. Cells were isolated from mice that express cGi500 under control of the SM22 α promoter and grown for 6 days. These primary VSMCs were imaged for cGMP followed by immunofluorescence staining for the contractile marker protein α SMA. Quantitative evaluation of fluorescence intensities (normalized to the mean of ANP-preferring cells) of individual VSMCs is shown. ANP-preferring cells and CNP-preferring cells are indicated by orange and cyan data points, respectively. Data are shown as mean + SEM (n = 95 cells of 141 cells on three coverslips from one cell isolation). Statistical significance is indicated by asterisks (***) p<.001).

5) Fig. 2 - it was unclear to me if “Data are shown as mean + SEM (n = 97-112 cells)” mentioned in the legend was only for 2f or all barplot panels.

We have now added n-numbers for every individual panel in all figures.

6) In contrast to the discrete classification of cells as ANP vs CNP preferring, is there a continuous gradient of cells in terms of preference, ranging from ANP to CNP. Since all cells not showing a strong preference either way are grouped into the ANP~CNP group, it is hard to assess whether that is the case or not. And if so, do other characteristics (e.g. morphological differences) also lie on a similar gradient? This could perhaps be assessed with the single cell measurements that the authors have already collected for these experiments.

Good point! We decided to classify our VSMCs in discrete groups to simplify data presentation. Along the reviewer’s comment, we have now analyzed if gradients exist for the “raw” preference values (ANP preference, ANP~CNP, CNP preference) as well as for morphology and aSMA expression.

Indeed, there is a continuous gradient concerning the natriuretic peptide preference of VSMCs as exemplified by re-analysis of the single-cell data shown in Fig. 1c (see figure for the reviewer, panel a). Furthermore, we analyzed a potential gradient for morphology and α SMA expression by plotting the “raw” preference values against the roundness and α SMA expression values, respectively, for each cell. While roundness and α SMA expression correlated significantly with natriuretic peptide preference, we did not see a clear gradient for either parameter (see figure for the reviewer, panels b, c). We have now included panel a of the figure for the reviewer as new Suppl Fig. 1a and discussed in the main text (page 7, 2nd para).

FIGURE FOR THE REVIEWER: Analysis of primary VSMCs for a continuous gradient regarding natriuretic peptide preference and for a correlation with roundness and α SMA expression. Several aspects of the primary VSMCs shown in Fig. 1 of the manuscript were analyzed using exact values for their natriuretic peptide preference instead of the categories ANP-, ANP~CNP, and CNP-preferring cells. **a**, Violin plot showing the distribution of natriuretic peptide preference by exact values. Classification in ANP-, ANP~CNP, and CNP-preferring cells is indicated by different colors as indicated to legend. **b**, Correlation of roundness and natriuretic peptide preference by exact values. Spearman’s rho: -0.42; $p < 1.7 \times 10^{-15}$. **c**, Correlation of α SMA expression and natriuretic peptide preference by exact values. Spearman’s rho: 0.64; $p < 6.1 \times 10^{-39}$. In all evaluations (a-c), ANP only and CNP only cells were excluded (“0” and “infinity”).

7) “Comparison of the top cell type-specific transcripts showed that the pSMC cluster was more similar to the mSMC cluster than to the cSMC cluster (Fig. 3b).” This needs to be backed by more quantitative inspection of the extent of similarity (e.g. cluster/ cell type level correlations).

We performed hierarchical clustering of the cells, based on the expression of the 3000 most variable genes (normalized and gene-wise scaled). Clustering was based on the Euclidean distance and the Ward’s minimum variance method for cluster distinction. Every cell in the resulting dendrogram was colored according to the clusters shown in Fig. 3. This approach

supports the conclusion that the pSMC cluster was more similar to the mSMC cluster than to the cSMC cluster (see figure for the reviewer).

Since this statement is not essential for our study and we do not want to add to the length of the manuscript by integrating further data (shown in the figure for the reviewer), we have deleted the sentence quoted above (page 9, 2nd para).

FIGURE FOR THE REVIEWER: Inspection of cluster similarity. Dendrogram showing the distance between cells/clusters based on hierarchical cluster analysis of the expression of the 3000 most variable genes (normalized and gene-wise scaled).

8) “Indeed, ≈24 % of the mSMCs co-expressed Npr2 and Tnfrsf11b mRNA (Fig. 3e yellow dots).” Could the authors please provide more details on how co-expression was defined in the single cell analysis?

Cells in which at least one transcript of both genes was detected were classified as co-expressing cells. We have now included this information in the legend to Fig. 3e). We are aware that this only defines the minimum number of co-expressing cells, as single-cell RNA sequencing is known to capture only a fraction of a cell's transcriptome.

9) In Fig. 4c,d the total cell counts are quite low, could the authors perform a statistical test (e.g. randomization test) to assess if the compositional changes reported are statistically significant?

In these experiments, we evaluated approximately 100 to 200 cells per condition (exact cell numbers are given in the figure legend), so not that few cells. We used a chi-square test to compare the distribution of VSMCs between the three categories (ANP, ANP~CNP, CNP). The changes in composition are statistically significant. This is now indicated in the updated figure.

10) “...only the outer cell layers of an atherosclerotic plaque, but not the inner core region, could be measured with sufficient intensity to evaluate cGMP signals...” Could the authors please comment on the implications of this limitation on their results?

Due to limitations in the penetration depth of light, we could only measure the outer layers of atherosclerotic plaques. These measured regions were most likely the fibrous cap, shoulder, and upper layers of the plaque core. However, for technical reasons, it was difficult to clearly assign a recorded FRET/cGMP signal to one of these specific plaque regions. To clarify this, we modified the respective text from “*However, only the outer cell layers of an atherosclerotic plaque, but not the inner core region, could be measured with sufficient intensity to evaluate cGMP signals*” (page 12, 1st para) to “*However, only the outer cell layers of an atherosclerotic plaque (presumably the fibrous cap, shoulder, and upper region of the plaque core), but not the inner core region, could be measured with sufficient intensity to evaluate cGMP signals*”.

The technical limitations of optical imaging described above meant that we were unable to characterize cells in the inner core of the plaque by cGMP/FRET imaging. However, please note that we also used the genetic GC-B lacZ reporter as an alternative method to detect GC-B-expressing plaque cells. This approach did not have the aforementioned limitations of *ex vivo* cGMP/FRET imaging of atherosclerotic lesions and clearly showed that cells with high GC-B expression exist in the plaque core (Fig. 6g, i). We added a respective statement to the discussion section (page 17, end of 1st para).

REVIEWER COMMENTS

Reviewer #1 (Remarks to the Author):

The authors have performed additional experiments to provide insight into how NP preference may affect VSMC state, which has increased the general impact of the study.

However, while the transparency about how the data was generated and selected has been increased in the revised manuscript (which allows evaluation of the findings), unfortunately, the information provided raises some concern about the robustness of data underlying key conclusions of the manuscript (examples below). This is important as the study builds on the *in vitro* findings. It will therefore be important to replicate experiments to provide information about variability between cells from different animals.

- It is not clear whether the statistical analysis takes into account that cells/regions from the same prep/animal are not independent (e.g. figure 4c/d + figure 7a/c/e/g).
- The data shown in figures 1b/c/d + 2b/d/f are from a single cell preparation.
- Figure 4b, supplemental 3c: data shown is from one isolation, - the result of an independent experiment could be shown as supplemental information.
- Figure 4c/d were from 3 cell isolations, but the variability between cells from different animals is not indicated.
- Information on replication is not provided for some of the supplemental figures.
- Data from a significant proportion of cells (15-35%) is discarded – while we understand this could be due to the technical aspects of the assay, it is important to rule out that this is not introducing bias, by replicating findings.

Minor points

We suggest including the analysis of human scRNA-seq datasets (shown as figure for the reviewer) in the manuscript to provide information about relevance in human disease – and consider including other cGMP pathway components.

We still do not find support for the claim on " page 14, 1st para", although we agree that this is suggested by the data (rather than "shown").

There is no evidence that the experiment in Figure 6h is sufficiently powered to test the null hypothesis that there is not a difference in one group (rebuttal point 7), which is needed to make this point.

Reviewer #2 (Remarks to the Author):

The authors have addressed all my comments satisfactorily.

Reviewer #3 (Remarks to the Author):

The authors have sufficiently addressed my comments. Congratulations on this beautiful work!

Reviewer #4 (Remarks to the Author):

Response to Reviewer #1

The authors have performed additional experiments to provide insight into how NP preference may affect VSMC state, which has increased the general impact of the study.

1) However, while the transparency about how the data was generate and selected has been increased in the revised manuscript (which allows evaluation of the findings), unfortunately, the information provided raises some concern about the robustness of data underlying key conclusions of the manuscript (examples below). This is important as the study builds on the in vitro findings. It will therefore be important to replicate experiments to provide information about variability between cells from different animals.

The in vitro data on ANP/CNP preference and cell phenotype were recorded in individual ~~primary VSMCs isolated from~~ mouse aortae. We are aware of the possibility of some variability between cells from different animals, cell isolations, or measurements. However, the main conclusion that phenotypic modulation of contractile VSMCs is associated with a switch from ANP/GC-A to CNP/GC-B signaling comes from a very high number of individual VSMCs that were measured in multiple cell isolations obtained from multiple mice with ubiquitous expression of cGi500 (cGi500-L1 mice). In these experiments, we recorded 2622 cells and analyzed 2440 cells. 7% of the recorded cells were excluded from analysis due to criteria that have been defined in the manuscript (e.g. due to poor quality of their ratio traces precluding classification of the cell). The cells were recorded on 45 coverslips derived from 13 independent cell isolations that have been prepared from 43 mice. As described below, we have replicated all key findings in independent experiments with multiple cell isolations.

Note that a cell isolation was typically NOT obtained from a single aorta or mouse, but from multiple aortae from multiple mice (e.g., one VSMC culture was obtained from 3 aortae from 3 mice). This information is now included in the Methods section. Pooling of aortae was necessary to obtain a sufficient number of cells for the experiments, and it also provided a way to mitigate uncertainty in data interpretation due to potential variations between cells from different animals. In a few cases, we have analyzed VSMC cultures that were indeed prepared from a single aorta. As shown in **Figure 1 for the Reviewer**, the results were similar with cells derived from different aortae/animals.

We believe that the very high n-number of observations (“data points”) that can be obtained by single-cell measurements, pooling of cells from multiple aortae/mice in each cell isolation, replication of key experiments and validation of findings with alternative methods (cGMP imaging, immunofluorescence staining, scRNA-seq, lacZ reporter cells, Western blotting) provides a solid basis for our main conclusions.

FIGURE 1 FOR THE REVIEWER: Expression of αSMA in primary VSMCs prepared from two individual aortae. Cells were isolated from individual aortae of two SMC-specific cGMP sensor mice (mouse #1, mouse #2). Cells were imaged for cGMP followed by immunofluorescence staining for the contractile marker protein αSMA. Quantitative evaluation of fluorescence intensities (normalized to the mean of ANP-preferring cells) of individual VSMCs is shown. ANP-preferring cells and CNP-preferring cells are indicated by orange and cyan data points, respectively. Data is shown as mean + SEM. Statistical significance is indicated by asterisks (** $p < .01$, *** $p < .001$).

2) It is not clear whether the statistical analysis takes into account that cells/regions from the same prep/animal are not independent (e.g. figure 4c/d + figure 7a/c/e/g).

As noted above (response to point #1), each measured cell population (e.g., Fig 4c/d) was derived from multiple aortae/mice. Moreover, data from several independent cell preparations were pooled before analyzing them by Pearson’s chi-squared test. These strategies should mitigate the potential influence of dependencies between cells from the same prep/animal on our results.

In Fig. 7 we compared plaques/plaque sections between mice with different genotypes (control vs. GC-B smko). As these groups were independent of each other, no paired tests were performed. Each experimental group contained several mice/aortae, and each mouse/aorta contributed several plaques/plaque sections to the dataset. It is difficult to account statistically for the possible dependence of some data points within these datasets. Note that even plaques from the same mouse can be very different. We analyzed for each mouse a segment of the atherosclerotic aorta covering approx. 600 μm. This procedure assured that for each mouse different plaques from different regions of the aortic arch were included in the analysis, thereby accounting for the intra-animal variability of atherosclerosis. This type of evaluation and data composition is frequently seen in mouse atherosclerosis experiments.

3) The data shown in figures 1b/c/d + 2b/d/f are from a single cell preparation.

Data shown in Fig. 1b/c/d and Fig. 2b/d/f are from single cell preparations. However, the respective experiments were replicated multiple times and/or validated with alternative methods throughout the study (see also response to point #1).

For instance, the finding that VSMCs can respond to ANP, CNP, and/or NO (Fig. 1b) was verified in all cGMP/FRET measurements with primary VSMCs under standard conditions, see **Figure 2 for the Reviewer**. Note that every data point in this graph represents an individual measurement of one coverslip with an average of 52 recorded cells. The coverslips were from 19 independent cell preparations, each of them prepared from multiple mice/aortae. The data underlying this graph is the same as shown in Suppl. Fig. 3c.

The classification of cultured VSMCs with preference for ANP or CNP (Fig. 1c) is representative for 62 measurements (coverslips) from 19 independent cell isolations (see Suppl. Fig. 3c). This has already been stated in our previous point-by-point response to Reviewer 1 (point #1): “Note that the classification of cultured VSMCs with preference for ANP or CNP has been carried out in multiple independent experiments (e.g., shown in Figs. 1, 2, and 4; Suppl. Figs. 1, 3, and 4). In total, we have analyzed >2400 cells in 45 measurements from global cGi500 sensor mice and >370 cells in 17 measurements from SMC-specific cGi500 sensor mice (see Suppl. Fig. 3c)“.

cGMP/FRET data recorded in GC-B knockout VSMCs showed that CNP-induced cGMP signals are mediated by GC-B (Fig. 1d). These results were confirmed by another independent experiment with GC-B knockout VSMCs that were transfected with the cGi500 biosensor. This information is now included in the legend to Fig. 1d.

In general, immunofluorescence stainings were replicated in multiple cell preparations, each obtained from multiple mice/aortae. For example, quantitative analyses of immunofluorescence stainings for α SMA from 5 independent cell isolations are shown in Fig. 2b, Fig. 2d, Suppl. Fig. 3b, and Suppl. Fig. 4b. Quantification of immunofluorescence stainings for SM22 α from 3 independent cell isolations are shown in Fig. 2b and Suppl. Fig. 3b.

That increased CNP-dependent cGMP signaling is associated with decreased expression of α SMA (Fig. 2b) and SM22 α (Fig. 2b) and increased expression of PDGFR α (Fig. 2f) and S100A4 (Fig. 2d) was also seen in single-cell RNA sequencing experiments (Fig. 3).

Further experimental evidence strengthening our conclusions is provided in **Figure 3 for the Reviewer**. The graphs show additional data for α SMA and SM22 α expression derived from experiments not included in the manuscript.

FIGURE 2 FOR THE REVIEWER: Characterization of cGMP responses in primary VSMC cultures from global (cGi500, green) and SMC-specific (SMC-cGi500, red) cGMP sensor mice. Each data point represents the percentage of cells in an individual measurement of one coverslip with an average number of 52 recorded cells. Response rates to ANP, CNP, or NO are shown as well as the fraction of cells that were excluded due to poor quality of their ratio traces (not applicable, na). Data are shown as mean + SEM (cGi500: 2622 recorded cells on 45 coverslips from 13 cell isolations; SMC-cGi500: 627 recorded cells on 17 coverslips from 6 cell isolations). THIS FIGURE EXTENDS THE DATA OF FIG. 1B AND SUPPL. FIG. 3C.

FIGURE 3 FOR THE REVIEWER: Correlation of αSMA and SM22α expression and cGMP response pattern at the single-cell level. VSMCs were isolated from cGi500-L1 or SMC-cGi500 mice. Cells were imaged for cGMP followed by immunofluorescence staining for the contractile marker protein αSMA (a) or SM22α (b). Quantitative evaluation of fluorescence intensities (normalized to the mean of ANP-preferring cells) of individual VSMCs is shown. ANP-preferring cells and CNP-preferring cells are indicated by orange and cyan data points, respectively. Data is shown as mean + SEM. Panel a: n = 95 cells of 141 cells on three coverslips from one cell isolation (left), n = 366 cells of 451 cells on three coverslips from one cell isolation (middle), n = 91 cells of 111 cells on two coverslips from two cell isolations (right). Panel b: co-staining of cells shown in panel a, right. n = 92 cells of 111 cells on two coverslips from two cell isolations. Statistical significance is indicated by asterisks (***) p<.001). THIS FIGURE EXTENDS THE DATA OF FIG. 2 BY ADDITIONAL INDEPENDENT EXPERIMENTS.

4) Figure 4b, supplemental 3c: data shown is from one isolation, - the result of an independent experiment could be shown as supplemental information.

As requested by the reviewer, we show an independent experiment demonstrating an increase of X-gal+ cells over time in primary cell culture (**Figure 4 for the Reviewer**) to support our findings shown in Fig. 4b of the manuscript.

We are confused why the reviewer concluded that the data shown in Suppl. Fig. 3c comes from one cell isolation. As stated in the figure legend, the data shown in this panel is from 13 cell isolations (cGi500 cells) or 6 cell isolations (SMC-cGi500 cells).

FIGURE 4 FOR THE REVIEWER: Development of GC-B expression in primary VSMCs grown for 4-14 days in the presence of 10 % FCS. LacZ GC-B reporter cells with GC-B expression were identified by their X-gal-stained nuclei (X-gal+/GC-B high cells). Each data point represents the percentage of X-gal+/GC-B high cells in a random field of view. Data is shown as mean + SEM (n = 10 regions on two coverslips per time point from one cell isolation). Statistical significance is indicated by asterisks (* p<.05; *** p<.001). THIS FIGURE EXTENDS THE DATA FROM FIG. 4B BY AN ADDITIONAL INDEPENDENT EXPERIMENT.

5) Figure 4c/d were from 3 cell isolations, but the variability between cells from different animals is not indicated.

In Fig. 4c/d, data on the effect of serum concentration and passaging on natriuretic peptide preference were pooled for each condition from multiple cell isolations, each derived from multiple mice/aortae. **Figure 5 for the Reviewer** shows the data separately for each cell isolation. In general, the results were similar for different cell preparations, supporting the conclusions drawn from Fig. 4c/d. High serum concentrations as well as passaging promote the synthetic CNP-preferring VSMC phenotype.

FIGURE 5 FOR THE REVIEWER: Promotion of CNP-preferring cells by phenotypic modulation. a, Effect of serum concentration on ANP/CNP preference of primary VSMCs in individual cell isolations. Primary VSMCs were grown in cell culture medium supplemented with 0.1 %, 10 % or 20 % FCS for before cGMP/FRET measurements. Drugs were applied as shown in Fig. 1c. Bars indicate the fraction of cells in each category under each condition. The red dashed lines separate independent cell isolations. Each cell preparation was obtained from multiple mice/aortae. n-numbers – 0.1 % FCS_isolation1/2/3: n = 63/47/57 cells of 63/50/59 cells; 10 % FCS_isolation2/3: n = 49/47 cells of 51/47 cells; 20 % FCS_isolation2/3: n = 69/52 cells of 70/53 cells. **b,** Effect of passaging on ANP/CNP preference of VSMCs in individual cell isolations. Primary (p0) and passaged (p2, p4) VSMCs were grown in 10 % FCS and analyzed by cGMP/FRET measurements. Bars indicate the fraction of cells in each category under each condition. The red dashed line separates independent cell isolations. Each cell preparation was obtained from multiple mice/aortae. n-numbers – p0_isolation1/2: n = 125/159 cells of 127/164 cells; p2_isolation1/2: n = 128/86 cells of 131/101 cells; p4_isolation1/2: n = 88/35 cells of 96/71 cells. Statistically significant differences in the distribution of ANP-preferring, ANP~CNP, and CNP-preferring cells under different conditions are indicated by asterisks (* p<.05; ** p<.01; *** p<.001). THIS FIGURE ILLUSTRATES THE VARIABILITY BETWEEN INDEPENDENT CELL ISOLATIONS FOR THE DATA SHOWN IN FIG. 4C/D.

6) Information on replication is not provided for some of the supplemental figures.

We have checked all suppl. figures and included the missing information in the figure legends.

7) Data from a significant proportion of cells (15-35%) is discarded – while we understand this could be due to the technical aspects of the assay, it is important to rule out that this is not introducing bias, by replicating findings.

As stated above, we have replicated our key results multiple times. We would like to point out that the exclusion of cells was based on clear criteria that have been defined in the manuscript, e.g., due to poor quality of their ratio traces precluding classification of the cell. In other words, the natriuretic peptide preference of these cells could not be assessed and, therefore, we could not include these cells as data points in the graphs. It is unlikely that such a procedure would introduce bias.

As depicted in **Figure 2 for the Reviewer** (right bars labeled “na”), only 7% of cells from global cGMP sensor mice (green data points) had to be excluded from analysis. About 40% of the cells from SMC-specific sensor mice (red data points) had to be excluded due to a poor FRET signal quality. The reason for the relatively poor imaging performance of the latter cell cultures is not entirely clear but might be related to low sensor expression and very low signal-to-noise ratios. However, as shown in **Figure 2 for the Reviewer**, the principal findings with cells from SMC-specific sensor mice were reproduced with multiple cell isolations and were consistent with the results obtained with cells from global cGMP sensor mice, which were used for most experiments.

Minor points

8) We suggest including the analysis of human scRNA-seq datasets (shown as figure for the reviewer) in the manuscript to provide information about relevance in human disease – and consider including other cGMP pathway components.

As suggested by the reviewer, we have now included the additional scRNA-seq datasets in the manuscript (see **new Suppl. Fig. 9**). As the focus of this manuscript is natriuretic peptide-dependent cGMP signaling, we did not include other components of the cGMP pathway.

9) We still do not find support for the claim on " page 14, 1st para", although we agree that this is suggested by the data (rather than "shown").

We have changed “showed” to “suggested”.

10) There is no evidence that the experiment in Figure 6h is sufficiently powered to test the null hypothesis that there is not a difference in one group (rebuttal point 7), which is needed to make this point.

We calculated the power for the indicated comparison of X-gal+ cells in media (M) and plaques (P) from control mice. The power of this comparison was 34 %. We now mention this limitation in the figure legend.

Reviewer #2 (Remarks to the Author):

The authors have addressed all my comments satisfactorily.

Reviewer #3 (Remarks to the Author):

The authors have sufficiently addressed my comments. Congratulations on this beautiful work!

Reviewer #4 (Remarks to the Author):

REVIEWERS' COMMENTS

Reviewer #1 (Remarks to the Author):

I have no further comments, thank you.